# Validation of the TROPOMI/S5P Aerosol Layer Height using EARLINET lidars

Konstantinos Michailidis[1*], Maria-Elissavet Koukouli[1], Dimitris Balis[1], J. Pepijn Veefkind[2,3], Martin de Graaf[2], Lucia Mona[4,], Nikolaos Papagianopoulos[4], Gesolmina Pappalardo[4], Ioanna Tsikoudi[5,6], Vassilis Amiridis[5], Eleni Marinou[5], Anna Gialitaki[5], Rodanthi-Elissavet Mamouri[7,8], Argyro Nisantzi[7,8], Daniele Bortoli[9,10], Maria João Costa[9,10], Vanda Salgueiro[9,10], Alexandros Papayannis[11], Maria Mylonaki[11], Lucas Alados-Arboledas[12], Salvatore Romano[13], Maria Rita Perrone[13], Holger Baars[14]

[1]Laboratory of Atmospheric Physics, Physics Department, Aristotle University of Thessaloniki, Greece
[2]Royal Netherlands Meteorological Institute (KNMI), De Bilt, the Netherlands
[3]University of Technology Delft (TU Delft), Delft, 2628 CN, the Netherlands
[4]Consiglio Nazionale delle Ricerche – Istituto di Metodologie per l'Analisi Ambientale (CNR-IMAA), C. da S. Loja, Tito Scalo (PZ), Italy
[5]IAASARS National Observatory of Athens, Athens, Greece
[6]Department of Physics, Section of Environmental Physics-Meteorology, National and Kapodistrian University of Athens, GR-15784 Zografou, Greece
[7]Department of Civil Engineering and Geomatics, Cyprus University of Technology, Limassol, Cyprus
[8]ERATOSTHENES Center of Excellence, Limassol, Cyprus
[9]Earth Remote Sensing Laboratory (EaRSLab), University of Évora, Évora, 7000-671, Portugal
[10]Institute of Earth Sciences (ITC) and Department of Physics, University of Évora, Évora, 7000-671, Portugal
[11]Laser Remote Sensing Unit, Department of Physics, National and Technical University of Athens, Zografou, 15780, Greece
[12]Andalusian Institute for Earth System Research, Department of Applied Physics, University of Granada, Granada, 18071, Spain
[13]Consorzio Nazionale Interuniversitario per le Scienze Fisiche della Materia and Università del Salento, Lecce, Italy
[14]Leibniz Institute for Tropospheric Research, Leipzig, Germany

*Correspondence: Konstantinos Michailidis (komichai@physics.auth.gr)

## Abstract

The purpose of this study is to investigate the ability of the Sentinel-5P TROPOspheric Monitoring Instrument (TROPOMI) to derive accurate geometrical features of lofted aerosol layers, selecting as study area the Mediterranean Basin. Comparisons with ground-based correlative measurements constitute a key component in the validation of passive and active satellite aerosol products. For this purpose, we use ground-based observations from quality controlled lidar stations reporting to the European Aerosol Research Lidar Network (EARLINET). An optimal methodology for validation purposes has been developed and applied using the EARLINET optical profiles and TROPOMI aerosol products, aiming at the in-depth evaluation of the TROPOMI Aerosol Layer Height (ALH) product for the period 2018 to 2022 over the Mediterranean Basin. Seven EARLINET stations were chosen, taking into consideration their proximity to the sea, which provided 63 coincident aerosol cases for the satellite retrievals. In the following, we present the first validation results for the TROPOMI/S5P ALH using the optimized EARLINET lidar products employing the automated validation chain designed for this purpose. The quantitative validation at pixels over the selected EARLINET stations illustrates that TROPOMI ALH product is consistent with the EARLINET lidar products, with a high correlation coefficient R=0.82 (R=0.51) and a mean bias of -0.51±0.77 km (-2.27±1.17 km) over ocean and land respectively. Overall, it appears that aerosol layer altitudes retrieved from TROPOMI are systematically

lower than altitudes from the lidar retrievals. High-albedo scenes as well as low aerosol load scenes are the most challenging for the TROPOMI retrieval algorithm, and these results testify to the need to further investigate the underlying cause. This work provides a clear indication that the TROPOMI ALH product can achieve, under conditions, the required threshold accuracy and precision requirements of 1 km, especially when only ocean pixels are included in the comparison analysis. Furthermore, we describe and analyse three case studies in detail, one dust and two smoke episodes, in order to illustrate the strengths and limitations of the TROPOMI ALH product and demonstrate the presented validation methodology. The present analysis provides important additions to the existing validation studies that have been performed so far for the TROPOMI S5P ALH product, which were based only on satellite-to-satellite comparisons.

# 1 Introduction

Aerosols play a key role in atmospheric composition, climate and human health (IPCC, 2021; Ramanathan et al., 2001). Given the broad variety of their natural and anthropogenic sources, their relatively short lifetime, and their different formation mechanisms, aerosols exhibit highly variable spatio-temporal distributions around the globe (Torres et al., 2007). Aerosol properties present one of the leading uncertainties in climate modeling; both natural and anthropogenic aerosols can strongly affect both air quality as well as the delicate balance in atmospheric chemistry (Bellouin et al., 2019; van Donkelaar et al., 2010). The knowledge of the vertical distribution on aerosols is an important key parameter to reduce uncertainties in our understanding of Earth's climate (Ramanathan and Carmichael, 2008). Accurate and reliable measurements of high spatio-temporal resolution aerosol distributions and their properties such as the aerosol layer height, ALH, is essential for understanding the impact of aerosols on the climate system.

Both active and passive remote sensing methods have been developed, from both ground-based and space-borne systems, in order to estimate the aerosol layer height. Ground-based active remote sensing methods offer high accuracy results however their geographical coverage is spatially limited. Space-based instruments are able to fill this gap, providing products with global coverage. In order to trust and use the space-based products, their validation against known ground truth is required. Lidar profiles from the European Aerosol Research Lidar Network, EARLINET, provide the accurate and reliable detailed vertical structure of the aerosols, and therefore can be regarded as the benchmark for validating passive ALH remote sensing (Pappalardo et al., 2014) . In recent years, many Earth satellite sensors have developed algorithms to extract the ALH information from their UltraViolet/Visible (UV/VIS) observations: the MetOp Global Ozone Monitoring Experiment-2 (GOME-2) instruments (Hassinen et al., 2016), the Deep Space Climate Observatory (DSCOVR) mission with its Earth Polychromatic Imaging Camera (EPIC) (Xu et al., 2019), the Multi-angle Imaging Spectro Radiometer (MISR) on board the NASA Terra satellite (Nelson et al., 2013), the Sentinel-5P TROPOspheric Monitoring Instrument (Veefkind et al., 2012) and more recently the Geostationary Environment Monitoring Spectrometer (GEMS; Kim et al., 2019). Over the next years,

upcoming missions such as), the Tropospheric Emissions: Monitoring Pollution mission (TEMPO) (Zoogman

et al., 2017) and the Sentinel-4 and Sentinel-5 missions (Ingmann et al., 2012) are expected to continue providing quality-assured aerosol height datasets.

In this work, we focus on the validation of the S5P/TROPOMI Aerosol Layer Height product (Nanda et al., 2020) against independent ground-based lidar measurements in order to enumerate the quality of the

90 TROPOMI retrievals. The EARLINET network of ground-based lidar instruments has been established to provide reference measurements of aerosol properties that can be applied to the validation of the TROPOMI retrievals, providing long-term, quality-assured and multi-wavelength aerosol vertical profiles. The geographic and temporal coverage of EARLINET stations alongside their quality assured measurements provides an excellent framework for the intercomparison of TROPOMI/S5P aerosol products under different

atmospheric conditions and aerosol concentrations around Europe. The ability of the EARLINET data to successfully assess and validate space-born ALH observations has already been demonstrated for the GOME2/MetOp aerosol height products (Michailidis et al., 2021).

Nanda et al. (2020) validated the TROPOMI operational ALH retrievals against the Cloud-Aerosol Lidar with Orthogonal Polarization CALIOP-based data, onboard the Cloud–Aerosol Lidar and Infrared Pathfinder

Satellite Observation (CALIPSO). Their work indicated that the operational algorithm retrieves lower ALH compared to CALIOP, by ~2 km over land and ~ 0.5 km over the ocean. The negative bias is primarily caused by the impact of the high surface reflectance in the $O_2$ A-band which affects aerosol retrievals. A similar comparison for the 2018 biomass burning fires in North America (Griffin et al., 2020) indicates that this bias also strongly depends on the thickness of the smoke plume. They reported a −2.1 km bias of ALH for thin

smoke plumes which is reduced to only~−0.7 km for plumes thicker than 1.5 km. At the time of writing this article, no studies assessing the validation of the TROPOMI aerosol height products with ground-based EARLINET stations have been published. A table in the end of the discussion section (Sect. 4) summarize the outcomes of this study, including the findings of the previous validation works.

The article is structured as follows: a general description of the region of interest is given in **Section 1.1.** **Sections 2.1** and **2.2** contain the description of the satellite and ground-based data sets used in the study. **Section 2.3** contains the detailed description of the validation methodology, the quality control and product limitations. In **Section 3** we provide the main validation results and statistics and also three case studies in order demonstrate the full potential of the presented method. Conclusions and prospects are summarized in

**Section 4**.

## 1.1   Study Region: the Mediterranean Region

The Mediterranean Sea Basin consists of a region heavily influenced by the Sahara Desert on the South and

120 the highly populated and industrialized European countries to the North. This region has been identified as a

crossroad of air masses with many types of aerosols (Lelieveld et al., 2002; Basart et al. 2009; Amiridis et al., 2010; Soupiona et al., 2020). This relatively high aerosol load in the region can have strong effects on the regional radiative budget, climate, and ecosystems (Stohl et al., 2015). Many studies have used satellite observations to derive aerosol properties over the Mediterranean during the last decade (e.g. Gerasopoulos et al. 2011; Mallet et al., 2013; Nabat et al., 2013; Marinou et al, 2017; Papanikolaou et al., 2020) and investigate their effect on radiation, cloud formation and climate (e.g. Georgoulias et al., 2020.)

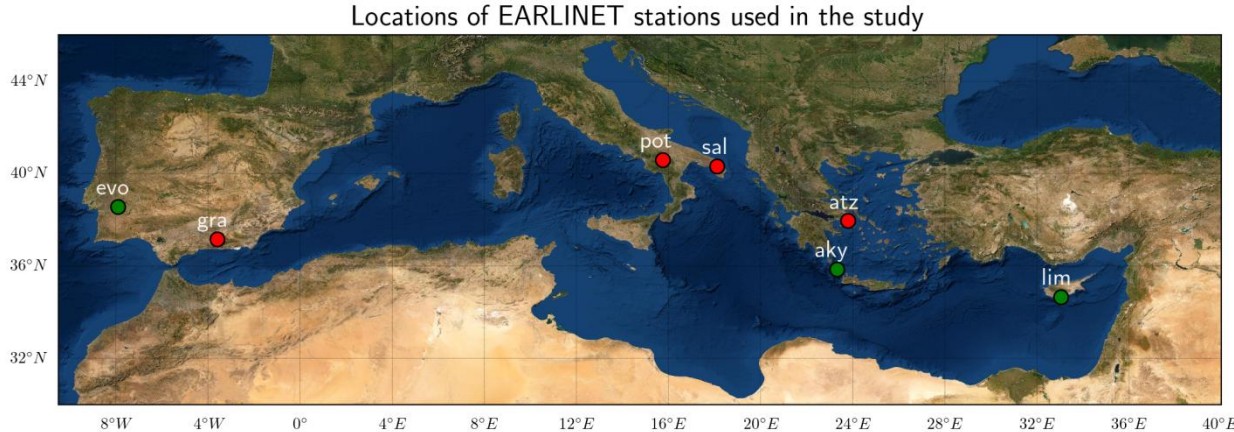

**Figure 1.** Location of the EARLINET lidar stations used in this study. Red circles denote multi-wavelength Raman lidars and green circles denote the stations with continuous operation capabilities (see Table 1).

Figure 1 shows the geographical distribution of the selected EARLINET stations participating in the TROPOMI-EARLINET intercomparison activity (in alphabetical order: Antikythera-PANGEA, Athens, Evora, Granada, Lecce, Limassol and Potenza). The list of stations along with their identification codes, surface elevation and respective references considered for the validation of the TROPOMI/S5P ALH product is shown in Table 1. The location of the stations across the Mediterranean basin is an ideal test environment for TROPOMI ALH features due to their proximity to the Sahara Desert and Europe, with frequently observed events of mineral dust and smoke particles (Lelieveld et al., 2002). Thence, the TROPOMI aerosol products can be examined under a complete set of different atmospheric conditions. Over land, the TROPOMI ALH product has decreased capabilities compared to over the sea surfaces since, over bright surfaces, the retrieval algorithm becomes increasingly sensitive to errors in the surface albedo features (Griffin et al., 2020; Sanders et al., 2015). Hence, the choice of the validation stations was performed based on the limitations of the TROPOMI ALH algorithm over land surfaces in providing accurate retrievals. Using these sites further provides an opportunity to study the effect of the albedo parameter on the validation of ALH product. A final aspect that influenced the choice of stations was the availability of suitable EARLINET data during the period examined, which is dictated by the fact that the EARLINET measurements are systematically performed following a standard schedule and not optimized for the validation of the TROPOMI products.

All participating stations operate high-performance multi-wavelength lidar systems. Three of the contributing stations (Antikythera-PANGEA, Evora and Limassol since 10.2021) are also part of the PollyNET sub-

network (http://polly.tropos.de, last access: 01 May 2022), operating 24/7 portable, remote-controlled multiwavelength-polarization Raman lidar systems (PollyXT; Baars et al., 2016; Engelmann et al., 2016).

Table 1. Details on the locations and main reference document for the EARLINET lidar stations used in this work.

| Station | Code | Country | Longitude, latitude, elevation | Main References |
|---|---|---|---|---|
| Antikythera-PANGEA | AKY | Greece | 23.31ºE, 35.86ºN, 193m | Kampouri et al. (2021) |
| Athens | ATZ | Greece | 23.78ºE, 37.96ºN, 212m | Pappayannis et al. (2020) |
| Évora | EVO | Portugal | 7.91ºW, 38.56ºN, 293m | Salgueiro et al. (2021) |
| Granada | GRA | Spain | 3.60ºW, 37.16ºN, 680m | Guerrero-Rascado et al. (2009) |
| Lecce | SAL | Italy | 18.10ºE, 40.33ºN, 30m | Perrone et al. (2019) |
| Limassol[1] | LIM | Cyprus | 33.04ºE, 34.67ºN, 10m | Nisantzi et al. (2015) |
| Limassol[2] | CYC | Cyprus | 33.03ºE, 34.67ºN, 11m | Mamouri et al. (2021) |
| Potenza | POT | Italy | 15.72º, 40.60ºN, 760m | Madonna et al. (2011) |

[1]Cyprus University of Technology (CUT)
[2]Leibniz Institute for Tropospheric Research, Leipzig and ERATOSTHENES Centre of Excellence (after Oct 2020)

## 2 Data and methodology

### 2.1 The EARLINET products

The lidar technique is the most predominant tool for aerosol profiling and has largely contributed to our knowledge of the vertical distribution of the aerosol optical properties (e.g., Balis et al., 2004; Papayannis et

al., 2008; Mona et al., 2012; Granados-Muñoz et al., 2016; Ortiz-Amezcua et al. 2017). The European Aerosol Research Lidar Network (https://www.earlinet.org/, Pappalardo et al., 2014), established in 2000, provides a large collection of quality-assured ground-based data of the vertical distribution of the aerosol optical and geometrical properties over Europe.

The EARLINET data have been used extensively for satellite aerosol products validation in recent years, such

as for the Cloud-Aerosol Lidar and Infrared Pathfinder Satellite Observations (CALIPSO) (Winker et al., 2009), which is the first satellite focused on monitoring vertically resolved aerosol and cloud optical products (Papagiannopoulos et al., 2016). Furthermore, the evaluation of aerosol optical products from the Cloud-Aerosol Transport System (CATS) on board the International Space Station (ISS) was also performed based on the EARLINET database (Proestakis et al., 2019). Recently, the co-polar particle backscatter coefficient

product measured by the Atmospheric LAser Doppler INstrument (ALADIN) onboard Aeolus, was evaluated in the Iberian Peninsula using EARLINET data (Abril-Gago et al., 2021). With respect to the aerosol layer height reported by UV-VIS satellite sensors, Michailidis et al., (2021) have successfully validated the GOME2/MetOp Absorbing aerosol height (AAH) products using aerosol profiles reported by the EARLINET community. The intercomparison showed that the GOME-2 AAH measurements provide a good estimation of

the aerosol layer altitudes sensed by the EARLINET ground-based lidars with a mean bias of approximately $-0.2\pm1.7$ km. While the TROPOMI ALH has more observations of dust and smoke outflows over the water surfaces, the GOME-2 AAH has improved availability over desert regions and remote oceans as its retrieval has no constraint on surface albedo and cloud fraction.

Currently, the network includes 32 active lidar stations distributed around Europe providing information of aerosol vertical distributions on a continental scale. The large majority of the stations involved is based on multi-wavelength Raman lidar systems, which combine detection channels at both elastic and Raman-shifted signals and are equipped with depolarization channels. Observations submitted to the EARLINET database follow absolute accuracy standards to achieve the desired confidence in product calculations. To this end, the lidar measurements are processed by the Single Calculus Chain (SCC) (D'Amico et al., 2015, 2016), the standardized tool that allows a centralized process of the lidar data acquired at each station within EARLINET. The SCC consists of several different modules for handling the pre-processing of raw lidar signals by applying specific corrections and providing the final optical products. In **Table 2**, the main Quality Assurance (QA) procedures, applied to the EARLINET lidars used in the study are presented. In order to make the lidar products from different systems in EARLINET comparable, and to be able to provide quality-assured datasets of network products, specific quality standards have been established (Freudenthaler et al., 2018) and algorithm intercomparison campaigns have been organized (e.g. Amodeo et al., 2018).

**Table 2.** QA procedures applied to EARLINET lidar measurements.

| Step | Procedure |
|:---:|:---:|
| 1 | **HiRELPP**: High Resolution EARLINET Lidar Pre-Processor<br>Corrections on the raw lidar signals before deriving higher level products.<br>(dead-time correction, trigger-delay correction, overlap correction, atmospheric and electronic background subtraction, low-and high-range automatic signal gluing) |
| 2 | **CloudScreen**: Cloud screen module<br>Clouds detection and screening on the pre-processed range corrected signal timeseries. |
| 3 | **ELPP**: EARLINET Lidar Pre-Processor<br>Corrections on the raw data before deriving the optical products at low temporal/spatial resolution. |
| 4 | **ELDA**: EARLINET Lidar Data Analyser<br>Retrieval of extinction and elastic/inelastic backscatter retrieval profiles |

The main information provided by the EARLINET database is the vertical distribution of aerosol backscatter and aerosol extinction coefficients alongside their errors, at one or more out of the following wavelengths: 355 nm, 532 nm and 1064 nm. The database also includes volume and particle depolarization ratio profiles at 532 nm (some stations also at 355 nm). During the daytime, the data acquisition is limited to the signals that occur from the elastic scattering of the laser beam by the air molecules and the atmospheric aerosol. The Klett–Fernald (KF) inversion is applied (Klett, 1981; Fernald, 1984), and the backscatter coefficient profiles are produced. In this study, only daytime lidar data from the QA EARLINET database were considered. A common source of uncertainty when dealing with lidar data is the system's overlap function that determines the altitude above which a profile contains trustworthy values. The incomplete overlap between the laser beam

and the receiver field of view significantly affects lidar observations of particle optical properties in the near-field range (first few hundred meters). The impact of the overlap height considered in the validation results is presented in detail in 2.3.

## 2.2 The TROPOMI/S5P Aerosol Layer Height

The TROPOspheric Monitoring Instrument (TROPOMI; Veefkind et al., 2012) is a space-borne, nadir-viewing, imaging spectrometer operating in a non-scanning push broom configuration covering wavelength bands between the ultraviolet and the shortwave infrared. Sentinel-5P is a near-polar sun-synchronous orbit satellite flying at an altitude of 817 km, with a 2600 km wide swath, providing near-daily global coverage and overpass local time at ascending node of 13:30 (repeat cycle of 17 days). The spatial resolution at nadir, originally of 3.5×7 km$^2$ (across-track × along-track) has been refined to 3.5×5.5 km$^2$ on 6 August 2019.

The TROPOMI ALH product focuses on the retrieval of vertically localized aerosol layers in the free troposphere, such as desert dust, biomass burning aerosol, or volcanic ash plumes. It can therefore provide accurate values to the modelling community by improving air quality forecasting and radiative forcing studies. The height of such layers is retrieved for cloud-free conditions and is reported in both altitude and pressure. The aerosol height retrieval is based on the absorption in the Oxygen-A band in the near-infrared wavelength range (759-770 nm) and assumes a single aerosol layer of 50 hPa thickness. This is an important simplification to note when comparing with other satellites and ground-based lidar profiles (e.g. from EARLINET), since these lidar profiles have the capability to detect multiple aerosol layers. The Oxygen-A band can provide altitude information on scattering layers (clouds or aerosol) from the troposphere up to the stratosphere. The TROPOMI Aerosol Layer Height (AER_LH) algorithm was developed by the Royal Netherlands Meteorological Institute (KNMI; Sanders et al., 2015; Nanda et al., 2018 Nanda et al., 2020) and is a part of the TROPOMI operational algorithm suite. We used S5P L2_AER_LH (RPRO & OFFL; Algorithm versions: 01.03.00 to 02.03.01) data covering the time period from June 2018 till July 2022. In brief, several quality control filters are applied in the TROPOMI L2 dataset, following the filtering proposed for ALH product (Nanda et al. 2020; their Table 1). Detailed description of the product, and product versions, can be found in the Product User Manual (PUM; Apituley et al., 2021).

Additionally, we use the operational UV Aerosol Index (UVAI) TROPOMI product (Stein-Zweers et al., 2021) to qualitatively examine the AER_LH products over the selected domain. The UVAI is an air quality product derived from the Top-Of-the-Atmosphere, TOA, reflectance spectra and is widely used as an indicator for the presence of aerosols in the atmosphere (e.g. Herman et al. 1997). The UVAI is based on spectral contrast in the UV spectral range for a given wavelength pair, where the difference between the observed

reflectance and the modelled clear-sky reflectance results in a residual value. Positive values indicate the presence of absorbing aerosols, such as dust, smoke, or volcanic ash. Over the oceans, positive UVAI may also result from non-aerosol sources such as sunglint and ocean color effects (Torres et al., 2018.) Clouds yield near-zero residual values and negative residual values can be indicative of the presence of non-absorbing aerosols, as shown by sensitivity studies of the UVAI (e.g. de Graaf et al., 2005). Negative UVAI can also result from optically thin clouds and aerosols over both land and oceans, while ocean color effects associated with chlorophyll absorption yield negative values over the oceans. For the aforementioned reasons, while the UVAI is well accepted as a first indicator of the presence of aerosols, care is required as to its quantitative interpretation. The use of this atmospheric parameter in this work is explained further in the results section.

## 2.3   Validation methodology and collocation criteria

In this section, we present in detail the basic principles of the validation method of the TROPOMI ALH product. The methodology is demonstrated using a selected number of collocated cases of TROPOMI overpasses over selected EARLINET lidar stations mainly located around the Mediterranean Basin for the period June 2018 to July 2022. The approach followed is based on the previous expertise and methodology that have been developed using EARLINET observations for the GOME2/MetOp validation activities (Michailidis et al., 2021). At present, seven EARLINET stations operating at 1064 nm (or 532 nm) channel contribute to this study (**Figure 1**). To obtain a validation dataset with statistical significance, ground-based lidar measurements first need to be collected and collocated with TROPOMI observations. Individual TROPOMI pixels are averaged over a selected radius around the lidar stations. Taking into account the recommendations of the previous comparison studies (Griffin et al., 2020; Nanda et al., 2020; Chen et al., 2021), the selection of data and the comparison between TROPOMI and EARLINET aerosol heights proceed through the following steps for each date:

1. Create a list of TROPOMI overpass swaths that are within the region of interest (EARLINET stations).

2. Identify the closest TROPOMI pixels (within a radius of 150km) around the EARLINET stations in time and space. A maximum time difference of ±4h is allowed between collocation pairs. This choice is a compromise to obtain a significant number of coincidences between two datasets. A shorter time and spatial coincidence criterion significantly decrease the numbers of sampled collocated days. In most of cases the time difference between the mean averaged lidar profiles and S5P overpass vary from 1 to 2 hours.

3. For each ground-based measurement, the spatially averaged TROPOMI pixels in a radius of 150 km, were selected for the comparison study. Different sensitivity tests have been performed in order to evaluate the robustness of the validation results. We used different radius around the EARLINET stations (from 50 to 150km). At most stations, the bias shows a small dependency on the radius.

4. As reference data input to the validation processing, we use the lidar backscatter coefficient profiles mainly at 1064 nm (or 532 nm), analyzed by the SCC (D'Amico et al., 2015; 2016) for quality-assured measurements and TROPOMI Level-2 ALH product. The backscatter coefficient profiles at 1064 nm is used for layer identification since the sensitivity to aerosol structures is higher at this wavelength than in the UV or visible

5. The TROPOMI/S5P pixel selection scheme and flags applied in the presented validation study were made following the recommendations on the Product Readme File (PRF) based on the quality assurance values (QA). In order to avoid misinterpretation of the data quality, we exclude satellite pixels associated with a "qa_value" below 0.5. This removes very cloudy scenes, snow-or ice-covered scenes and problematic retrievals. In addition, residual cirrus clouds can cause substantial retrieval biases. For our purposes, the

"cirrus_reflectance_viirs_filter" flag is the preferred flag for removing possibly cirrus cloudy pixels. Satellite pixels with VIIRS average cirrus reflectance beyond 0.4 are excluded from the analysis. A sun-glint mask is also applied to screen sun-glint regions. Pixels with an associated negative AI are excluded; hence only desert dust, biomass burning aerosol and volcanic ash aerosols – i.e. absorbing aerosols – remain in the dataset.

The effective aerosol height may be also described in terms of layer boundaries or by the full vertical profile. In this study, we make use of the backscatter-weighted height (ALH$_{bsc}$), calculated as the center of mass (Z$_{COM}$) on backscatter ($_{bsc}$) profile, based on the methodology described in Mona et al. (2006). This height parameter is an important indicator for vertical profiles that gives in a single number an indication of the

altitude of the aerosol distribution. For example, in cases where a single aerosol layer is present in the atmosphere, the ALH$_{bsc}$ gives an indication of its mean altitude; in case of multiple layers however, the ALH$_{bsc}$ could be located in areas without any considerable aerosol load. In addition, ALH$_{bsc}$ is considered ideal for comparisons with aerosol layer height retrievals from passive remote sensing (e.g. TROPOMI/S5P, GOME-2/MetOp and upcoming Sentinel-4 & 5 missions). Information about the aerosol layer center of mass

is useful because the characteristics of the detected layer can be distinguished at this altitude. Under the detection of a homogenous aerosol layer, the Z$_{com}$ can be estimated as the mean altitude of the identified aerosol layer weighted by the altitude-dependent aerosol backscatter coefficient. In some cases, the aerosol vertical structure is very complicated because aerosol layers are present at different heights. For these cases, a total layer resulting from the multi-layered structure is considered for the calculation of mean optical

parameters and integrated values. The weighted-backscatter altitude is estimated by the equation:

$$ALH_{bsc} = \frac{\int_{z_{i=1}}^{z=n} z_i \cdot \beta_{aer,i}(z)dz}{\int_{z_{i=1}}^{z=n} \beta_{aer,i}(z)dz} \qquad \text{(Equation 1)}$$

where $\beta_{aer,i}$ represents the aerosol backscatter coefficient (Mm$^{-1}$·sr$^{-1}$) primarily at 1064 nm channel at level i

and Z$_i$ is the altitude (km) of level i for the aerosol profile signal. Based on the above equation, the layer height

is calculated from backscatter profiles, symbolized as $ALH_{bsc}$. We found that there is no significant dependence on the choice of wavelength channel ranging between $20 - 80m$ in the ALH calculation. The $ALH_{bsc}$ represents an effective ALH weighted by the aerosol backscatter signal at each level and is the best parameter to compare with ALH as defined in the TROPOMI algorithm. In our analysis, we applied Equation 1 to all lidar backscatter profiles collocated to TROPOMI measurements. The backscatter profiles are used from each station together with the associated error in the vertical profile. After applying the Monte Carlo error propagation using the backscatter profiles and the errors, for all the cases, we found that the effect on the estimated $ALH_{bsc}$ is small, i.e. of the order of some tens of meters, ranging between 10 - 60 m.

A very important and critical issue in lidar-based ALH calculation is the incomplete overlap area between the laser beam and the receiver field of view. This affects the observations of the optical properties of the particles in the first hundreds of meters. To overcome this issue, we rely on certain assumptions. To calculate the $ALH_{bsc}$ from the lidar backscatter profiles using equation 1, for the height range between the surface and the full overlap height, we assumed a constant backscatter coefficient (height-independent) equal to the one measured at the full overlap height. This is generally acceptable since the Planetary Boundary Layer (PBL) is characterized by well-mixed aerosol conditions (Siomos et al., 2018). We can conclude that the effect of this assumption shown on the calculation of $ALH_{bsc}$ is of the order of 100 - 400m, depending on the technical characteristics of the lidar systems. Furthermore, we also investigated the effect of lidar ratio on the $ALH_{bsc}$ estimates based on lidar backscatter profile retrievals for different lidar ratio values. The findings show that the effect of the on the weighted height calculation ($ALH_{bsc}$) is small, lower than 40m.

Following the work proposed by Michailidis et al. (2021) for automatic layer detection, we also apply the WCT (Wavelet Covariance Transform) approach in order to check whether the TROPOMI retrieved ALH is sensitive to distinct layers rather than a representative effective layer from the whole profile. In this methodology all possible individual layers identified by the lidar observations are analysed autonomously, providing individual assessments on the height of the aerosol mass, and not a mean effective height from all layers as is extracted by Eq. 1. Hence, further below, a comparison is also given using the WCT formalism on the lidar profiles, from which the layer considered most optically significant is compared against TROPOMI ALH retrievals. In the case that more than one layers with a significant contribution to the optical thickness of the profile exist, an average value between these is calculated for the comparison against satellite height retrievals.

## 3  Validation Results and Discussion

Following the methodology summarized in **Section 2.3**, we performed a validation analysis using lidar data from 7 ground based EARLINET sites located across the Mediterranean, spatio-temporally collected with data from TROPOMI instrument aboard Sentinel-5P satellite. The results of this analysis are presented and

discussed in detail in **Section 3.1**. Furthermore, in **Section 3.2**, three selected representative cases are presented in detail, during extensive dust and smoke events over the Mediterranean Basin in order to illustrate strengths and limitations of TROPOMI ALH product.

## 3.1 Comparison of TROPOMI against EARLINET ALH

Here we present the first comparison between TROPOMI ALH and EARLINET measurements over the Mediterranean. The basic issue in this validation approach was the difficulty in identifying good

spatiotemporal collocations between EARLINET lidar stations observations and TROPOMI/S5P overpasses. The TROPOMI AER_ALH retrievals over land surfaces is a challenge and strongly dependent on the surface albedo, with low accuracy over bright surfaces. First comparison results presented here confirm this feature. Overall, from the selected EARLINET stations across the Mediterranean, 63 coincident cases were found, checked and flagged for the comparison against TROPOMI retrievals. The collocated aerosol backscatter

profiles at 1064nm from lidar level-2 products are used to calculate a $ALH_{bsc}$ for the validation of TROPOMI ALH. The spatially averaged TROPOMI ALH retrievals in a radius of 150km around the station, around the overpass time of TROPOMI are used in the validation for each day. The total available dataset is on the small side but suitable for the comparison study and general representativeness of the TROPOMI ALH product. For each selected satellite file, TROPOMI UVAI data were used to qualitatively discriminate aerosol plumes from

the background. We apply our validation process using satellite retrievals separately over land and water surfaces to further demonstrate the known TROPOMI ALH issues over land. The surface reflectance for each pixel is derived after classifying the land and water surface based on the pixel location. Recall that over land the TROPOMI ALH product has decreased detection capabilities than over the sea surfaces since, over bright surfaces, the retrieval algorithm becomes increasingly sensitive to errors in the surface albedo features

(Sanders et al., 2015).

**Figure 2** shows the scatterplot of TROPOMI ALH against EARLINET $ALH_{bsc}$ for all the common cases used for the intercomparison. Ocean-only S5P pixel comparisons are shown in **Figure 2** (upper) and only land pixels are shown in **Figure 2** (middle). The error bars represent the corresponding spatial standard deviation of

375 TROPOMI pixels within 150km of the EARLINET sites. The colour scale indicates the averaged TROPOMI aerosol index values. The agreement between the TROPOMI-EARLINET datasets was quantified by several evaluation metrics, including the number of collocations (N), the linear correlation coefficient (R), the slope (a) and intercept (b) of the linear regression, the root mean square error (RMSE), the mean of the absolute (absolute bias) and relative (relative bias) for all cases. By defining a weighted height from EARLINET

aerosol backscatter profile products ($ALH_{bsc}$), the quantitative validation at pixels over the selected EARLINET stations illustrates that TROPOMI ALH is consistent with $ALH_{bsc}$, with a high correlation coefficient R=0.81 and mean bias -0.51±0.77 km over ocean pixels and R=0.51 and -2.28±1.17 km over both land pixels, respectively.

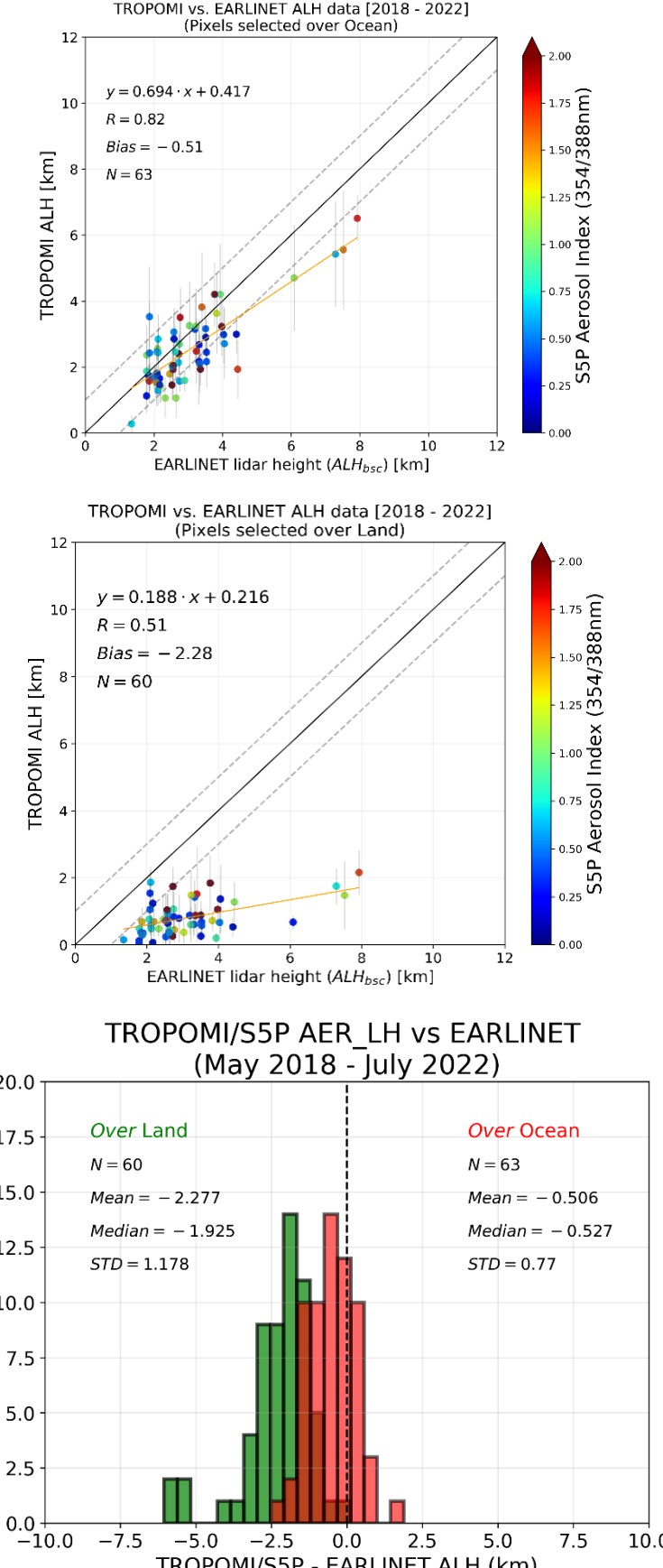

**Figure 2.** Scatterplots of TROPOMI against EARLINET data: (upper) TROPOMI pixels over ocean and (middle) over land. The color of each scatter point indicates the TROPOMI retrieved UVAI values, and the error bars of each scatter

indicate the spatial variability of the averaged TROPOMI ALH pixels. (Lower) Histogram of the differences between TROPOMI and EARLINET datasets, in red over the ocean pixels and in green for both ocean and land pixels.

From these scatter plots, it can be noted that there are two different point clusters which represent two different aerosol events. The cluster with the low aerosol layer heights represents the dust episodes, while the high aerosol loads represent the four collocated cases associated to smoke events over the west, central and east Mediterranean which originated from the California forest fires. These cases have thin and well defined layer structures, with no significant contribution from lower layers, apart from the PBL structure. Overall, the TROPOMI ALH retrievals are systematically lower than the compared lidar height in both clusters. This can be seen also in **Figure** *2* (lower) which presents histogram plots of absolute differences. The magnitude of the mean height difference is smallest when only ocean pixels (in red) are included in the comparison with the EARLINET and increases when compared with land pixels (in green). Many factors can play a role in this apparent disagreement between TROPOMI retrievals over land and sea including that high surface albedos negatively influence the ALH, biasing the ALH towards the surface. The accompanied related statistic metrics are summarized in Table 3. The main reason for the strong underestimation of the aerosol layer height retrieved by the current algorithm from TROPOMI over land is the surface reflectivity climatology used in the forward model, leading in biased or non-convergent retrievals over land. Sensitivity studies showed that the observed large bias over land is reduced when fitting of the surface albedo as estimated from TROPOMI itself was included in the retrieval procedure. This will be further investigated in the near future and is intended to be implemented in future versions of the ALH L2 product.

Table 3**.** Statistics of the comparison between TROPOMI and EARLINET ALH datasets.

| TROPOMI pixels | $N^a$ | $R^b$ | Slope$^c$ | $Y^d$ | $MB^e$ | $RB^f$ | $RMSE^g$ |
|---|---|---|---|---|---|---|---|
| Ocean | 63 | 0.82 | 0.25 | -0.41 | -0.51±0.77 km | -14.73% | 0.9 km |
| Land | 60 | 0.51 | 0.19 | 0.21 | -2.28±1.17 km | -73.36% | 2.55 km |

[a]Number of collocations,[b]Correlation coefficient, [c]Slope from linear regression fit , [d]Y-intercept of linear regression fit, [e]Mean bias, [f]Relative Bias, [g]Root mean square error

The mean maximum layer heights for all cases are, on average, 2.51±1.13 km (ranging between 0.27 and 6.5 km) and 0.791±0.5 km (ranging between 0.06 and 2.15 km) for TROPOMI ocean and land pixels, respectively. For the case of EARLINET data the mean layer height is 2.87±1.28 km (ranging between 1.16 and 7.22 km). We have to note again that for the calculation of the lidar ALH, the overlap effect and the station altitude are taking into account, following the assumptions discussed in 2. The above-mentioned statistics are summarized in Table 4.

Table 4**.** Layer height retrievals (min, max and average) of the EARLINET and TROPOMI ALH collocations.

| Instrument | Min height (km) | Max height (km) | Average height (km) |
|---|---|---|---|
| TROPOMI (ocean) | 0.27 | 6.5 | 2.51±1.13 |

| | | | |
|---|---|---|---|
| TROPOMI (land) | 0.06 | 2.15 | 0.791±0.5 |
| EARLINET | 1.16 | 7.22 | 2.87±1.28 |

The possibility of multiple atmospheric aerosol layers is a challenging feature for a passive sensor to retrieve, obviously affecting ALH. We hence performed a separate analysis for the cases that only one layer was detected and for the cases where two-or-more layers were detected by EARLINET. We apply the WCT approach on the backscatter profiles to distinct the aerosol layers in an automatic way for all the cases, as described in Michailidis et al., 2021. Overall, the lidar data reveal the presence of a single aerosol layer in 57.1% (N=36) of sample cases and a multilayer structure (two-or-more layers) the 42.9% (N=27) of the total sample cases. It was found that the mean bias of the ALH differences does not significantly vary (for ocean: -0.55±0.66km & land: -2.38±1.24km) when multilayers exist in the atmospheric scene, but we cannot say with certainty that it is the general rule due to the use of the limited validation dataset. Overall, among all the cases, the best performance of the TROPOMI ALH is recorded in cases of single well-developed dust layers. However, further research is needed to substantiate the observations and make conclusive quantitative statements. Using the WCT method for analysing the lidar profiles for validating purposes adds value in the case of multi-layer structures but to refine this automated technique, more cases have to be analysed in the future, as these become available.

Another factor that can affect the satellite-ground based intercomparison of measurements/products is the topography. Over areas with a complex terrain, vertical inconsistencies between ground-based and satellite retrievals may appear due to orography induced disturbances in the aerosol layer height. According to the statistics summarized in Table 5, the correlative measurements between the mountainous EARLINET stations (Potenza, Granada, Evora) and the S5P overpasses show similar mean biases as in the case of coastal stations (Limassol, Lecce, Antikythera, Athens). The complex topography, in terms of geographical characteristics, the horizontal distance between the TROPOMI retrieved pixels and the ground-based lidar sites are however features that should also be examined when inter-comparing EARLINET and TROPOMI aerosol layer heights.

Table 5. Clustering of EARLINET stations with respect to topographical features and the corresponded TROPOMI-EARLINET mean bias. Details about the EARLINET stations are provided in Table 1.

| | TROPOMI pixels over ocean | | TROPOMI pixels over land | |
|---|---|---|---|---|
| **Clusters** | **No of cases** | **MB±STD [km]** | **No of cases** | **MB±STD [km]** |
| **Coastal stations:** AKY, ATZ, SAL, LIM, CYC | 46 | -0.47±0.69 | 43 | -2.12±1.05 |
| **Mountainous stations:** POT, GRA, EVO | 17 | -0.61±0.91 | 17 | -2.67±1.33 |

## 3.2 Case studies: Analysis and results

Three typical days with sufficient aerosol load over the Mediterranean were selected to illustrate the performance of the TROPOMI ALH product over scenes with strong aerosol load. These cases refer to Antikythera, Evora and Potenza lidar observations during extended dust and smoke events. The selected cases include: (a) a Saharan dust outbreak over eastern Mediterranean region on 22[nd] of June 2021 and (b) a smoke aerosol plume transported during 4 days between the 24[th] and the 27[th] of October 2020, originating from the large wildfire episodes in the California region (N. America).

### 3.2.1 Dust case over Eastern Mediterranean: 22 June 2021, PANGEA observatory (Greece)

On the 22[nd] of June, the Eastern Mediterranean was affected by a strong dust episode originating from North Africa. On this day, the TROPOMI overpass over Greece was between ~10:00 and 11:00 UTC. A lofted layer of dust was also clearly observed by the PollyXT system at PANGEA Antikythera station on the same day. The PANGEA observatory of NOA on the remote island of Antikythera is located across the travel path of different air masses, providing continuous monitoring of essential climate variables in the Eastern Mediterranean (Kampouri et al., 2021). A PollyXT NOA lidar (Engelmann et al., 2016) is installed in the PANGEA observations in Antikythera since August 2018. This multi-wavelength system is part of the EARLINET community, with 24/7 operational capabilities, providing vertical distributions of aerosol properties at different wavelengths. The dust plume can clearly be seen in **Figure 3** (left), over Greece from the VIIRS/Suomi-NPP true color image for this day (VIIRS images are generated from https://wvs.earthdata.nasa.gov/; last access: 1 May 2022). With the red star symbol the PANGEA lidar station at Antikythera is depicted. **Figure 3** (center) and **Figure 3** (right) show the TROPOMI ALH and UVAI product retrievals, during the investigated dust episode. Comparing the TROPOMI product maps to the VIIRS image, it can be seen that the large positive UVAI pixels and elevated aerosol layers are located at the detected plumes. We note here that a recent publication addresses the treatment of clouds in the UVAI parameterization for OMI/Aura observations (Torres et al., 2018) and such considerations should be made in studies that depend numerically on the UVAI atmospheric parameter.

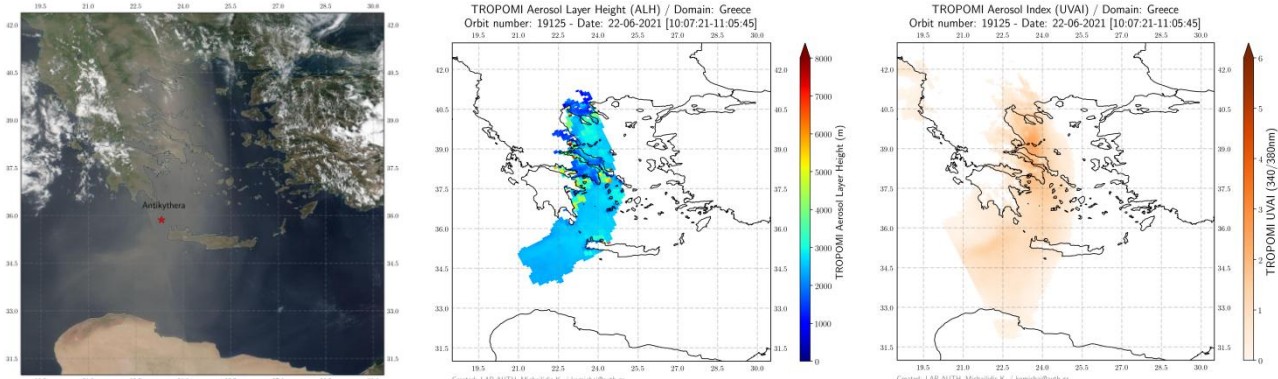

**Figure 3**. (Left) Suomi-NPP VIIRS True colour image on the 22[nd] June 2021. The red star indicates the position of the Antikythera lidar station. (Center) TROPOMI ALH and (right) TROPOMI UVAI over Greece.

**Figure 4** (left) presents the total attenuated backscatter signal time-series for the PollyXT at 1064nm, over Antikythera between 10:00-13:00 UTC on this day. A homogeneous layer can be identified throughout the

day below 5 km (**Figure 5**). The aerosol load is mainly between 500 and 4500 m and the sky above the site is cloud-free during the TROPOMI overpass time (thick red line). Additionally, in order to verify the origin of the detected aerosol layers, we calculated back-trajectories by using the HYSPLIT model (Hybrid Single-Particle Langrangian Integrated Trajectory; available online: last access: 01 May 2022; Stein et al. 2015). The temporal evolution of 3-day backward trajectories, for the 22$^{nd}$ of June, for selected arrival heights (2000m (red), 3500m (green) and 5000m (yellow)) is illustrated in **Figure 4** (right). As can be seen, the air masses arrived over Antikythera station follow a pattern originating from North-western Africa.

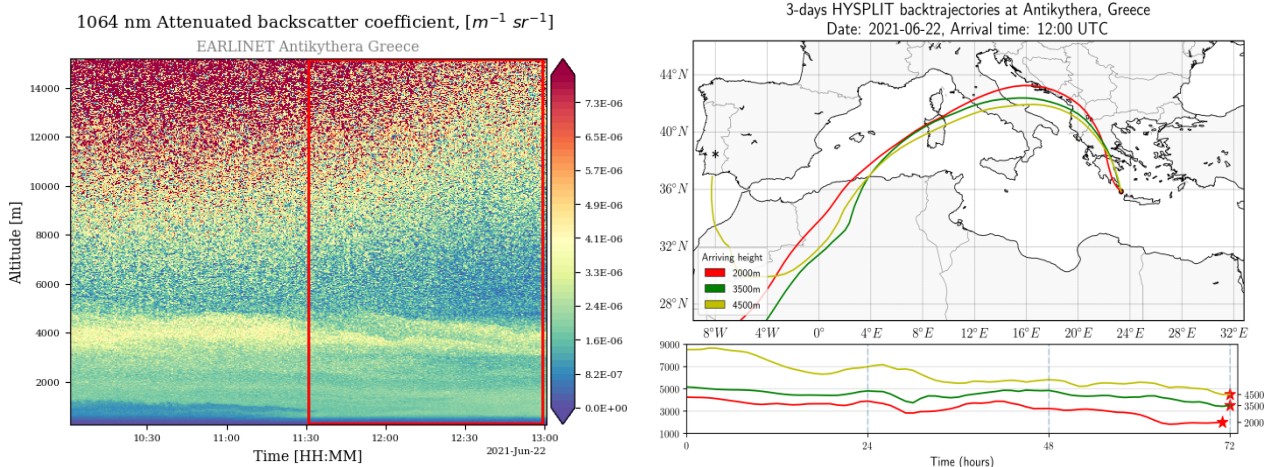

**Figure 4.** (Left) Temporal evolution of the total attenuated backscatter signal from the PANGEA PollyXT at 1064nm and (right) 3-day back-trajectories arriving at Antikythera, Greece on 22 June 2021 at 12:00 UTC (HYSPLIT accessible at www.ready.noaa.gov).

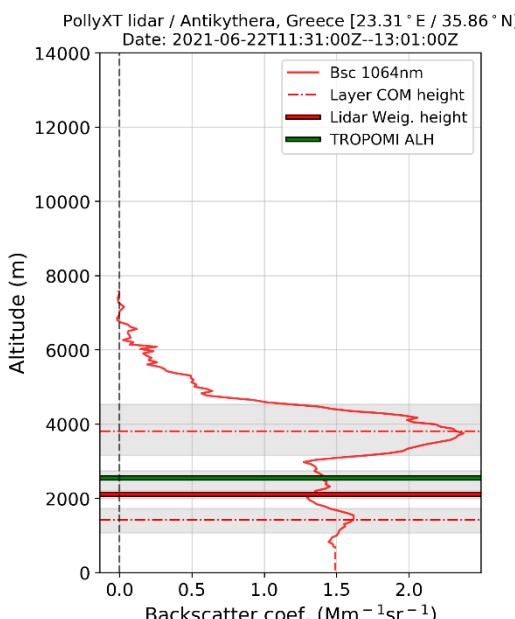

**Figure 5**. Lidar backscatter profile at 1064nm (ALH$_{bsc}$, red line) at Antikythera, Greece on 22 June 2021. The Lidar weighted aerosol height by Eq. 1 is shown as a thick red line. The calculated center-of-mass (COM) of the two identified layers are shown as dash-dot red lines and their thickness as the grey areas. The TROPOMI mean ALH is given as the thick green line.

For this event, the TROPOMI ALH spatially averaged values and the EARLINET temporally averaged backscatter coefficient profiles (between 11:30-13:00 UTC) are qualitatively compared in **Figure 5**. Two optically thin layers with a thickness of less than 300 m were detected, with centers of mass shown as the red dash-dot lines and their spread as the grey areas. TROPOMI detects this layer at 2550m while the calculated $ALH_{bsc}$ by applying Eq. 1 to the lidar profiles places it at 2105m. An agreement within 500m between the satellite and ground-based lidar systems is hence found for this clear aerosol scene, within the target requirement for the TROPOMI ALH product (ATBD; de Graaf et al., 2021). The presented case study indicates that, under cloud-free conditions where homogeneous aerosol layers are developed, the mean ALH value retrieved by the TROPOMI is in satisfactory agreement with the calculated $ALH_{bsc}$ from the lidar profile, confirming the findings of Griffin et al. (2020) and Nanda et al. (2020).

Using the WCT algorithm on the $\beta_{aer}$ profile at 1064 nm, we can also extract the geometrical properties of detected layers. Once the top and the base of the aerosol layer are identified, the $Z_{com}$ and optical properties of each aerosol layer can be also estimated. In this way, we can also investigate if there are strong variations in the $b_{aer}$ (or $_{bsc}$), which may lead to an identification of a separate layer. For the predominant thick layer (upper layer shown in **Figure 5**), the retrieved geometric properties are: layer base (3150 m), layer top (4350 m), layer thickness (2200 m), layer center of mass (4000 m). This example amply demonstrates that when using the WCT technique as a reference, in the presence of two layers with different spreads, the best agreement with the satellite estimate is not necessarily found for the optically thickest one. The representativeness of the TROPOMI ALH when multiple layers are present is undoubtedly an issue of further investigation in the future.

### 3.2.2   Smoke advection over the Mediterranean from Californian fires

In mid-October 2020, a series of wildfires took place in Northern California resulting in thousands of square kilometers of boreal forest being burned and causing a huge amount of smoke to enter the atmosphere. The emissions caused extreme air pollution conditions with poor visibility throughout the area for several days. The TROPOMI sensor has been monitoring these wildfires, and tracked the smoke as it travelled all the way across North America and the Atlantic Ocean to arrive in Mediterranean (Baars et al., 2021; Ansmann et al., 2021). These smoke aerosol layers were transported from the US west coast towards Europe within 4-5 days. The smoke arrived over the Iberian Peninsula in southwestern Europe on 24 October (**Figure 6**a), just in time for a regular overpass of the TROPOMI over Iberian Peninsula. As the plume was transported along the Mediterranean, it was detected over southern Italy and Greece, shown in the true color images from VIIRS/Suomi-NPP (**Figure 6**b to d). The S5P trails behind Suomi-NPP by 3.5 min in Local Time Ascending Node, allowing its swath to remain within the scene observed by Suomi-NPP. We have to note here that the ALH can be very sensitive to cloud contamination as aerosols and clouds can be difficult to distinguish. In general, the VIIRS cloud mask has good performance for pixels covered by aerosol plumes, but in many cases

where very thick layers are detected the cloud mask can misclassify the retrievals pixels as cloudy pixels with
high cloud fraction. The equivalent daily TROPOMI UVAI and ALH product retrievals are presented in
**Figure 7** and **Figure 8**, respectively. The detected smoke plumes are highlighted by large positive values of
UVAI, which are in contrast to clouds that typically exhibit a negative UVAI or/and close to zero (Torres et al.,
1998).

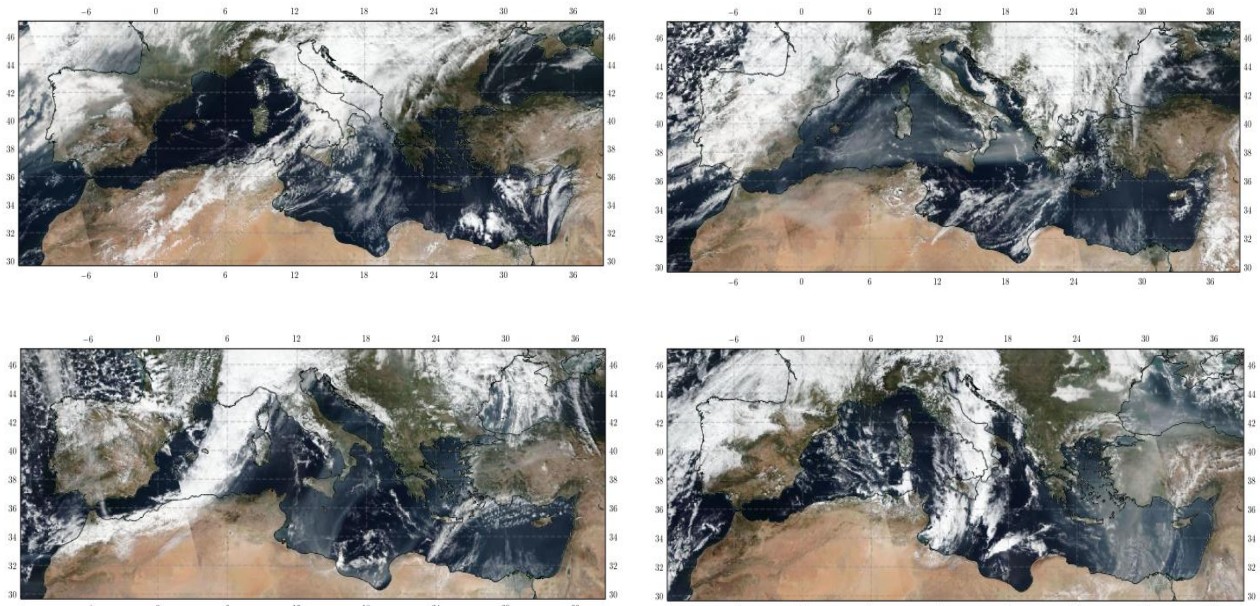

**Figure 6**. VIIRS/Suomi NPP True color images for the four smoke scenes on 24-27 October 2020 (Maps are generated
from NASA Worldview Snapshots: **https://wvs.earthdata.nasa.gov/**)

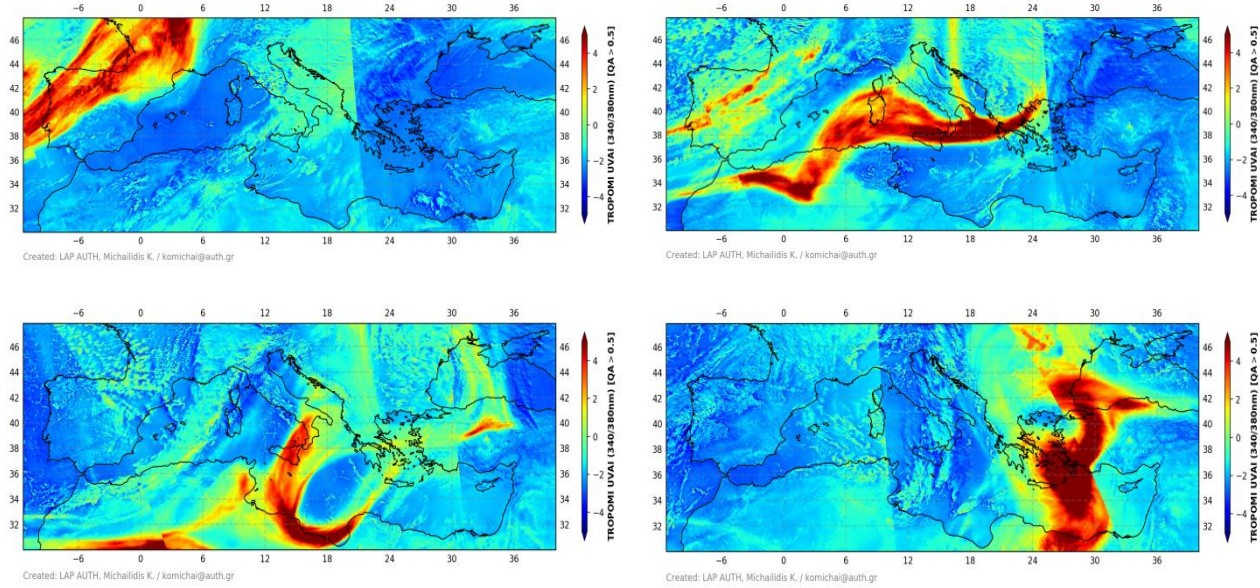

**Figure 7**. TROPOMI UV aerosol index (UVAI) retrievals during the smoke plume transport over the Mediterranean,
during 24-27 October 2020.

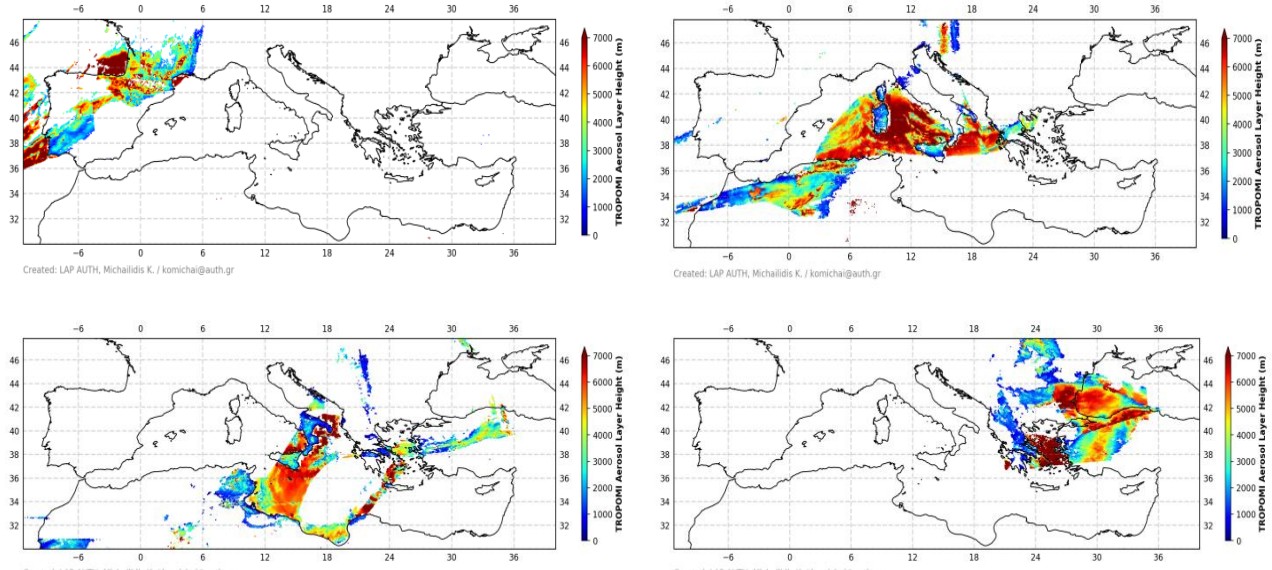

**Figure 8.** TROPOMI Aerosol Layer Height (ALH) retrievals during the smoke plume transport over the Mediterranean, during 24-27 October 2020.

- **Smoke case over western Mediterranean: 24 October 2020, Évora (Portugal)**

On 24 October 2020, during the Suomi-NPP satellite overpass, the VIIRS sensor captured the true color image (**Figure 9**, left) showing a large amount of smoke plumes in the western half of the Iberian Peninsula , particularly in the north-western areas facing the Atlantic Ocean. The selected scene is also strongly affected by the presence of clouds. TROPOMI overpassed Évora around 12:30 UTC on this day and recorded very high values related to UVAI values (>5) (**Figure 9**, right) as well as elevated aerosol plumes corresponding to high ALH values (**Figure 9**, center). The maximum altitude in the TROPOMI AER_LH data is about ~8 km. The white spaces in the TROPOMI product maps indicate no valid TROPOMI retrievals over these areas.

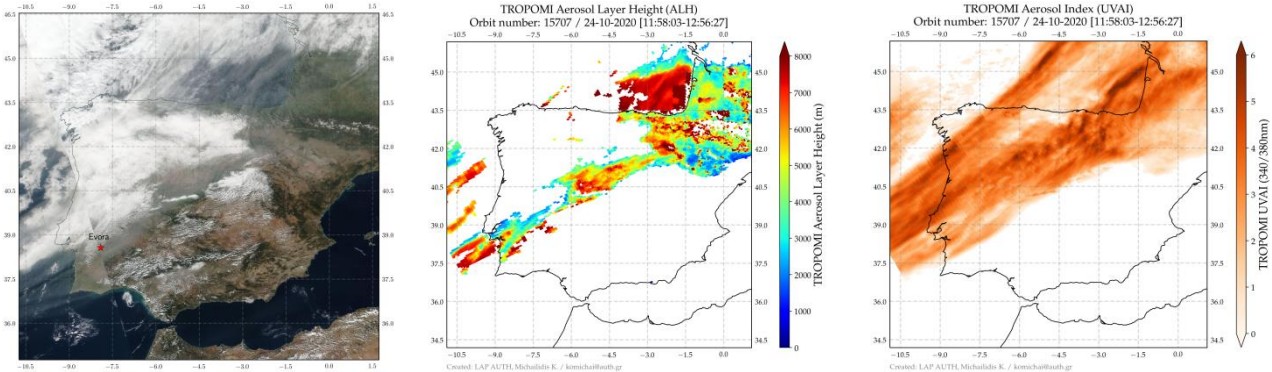

**Figure 9.** (Left)VIIRS Suomi-NPP true colour image on 24 October 2020 over the Iberian Peninsula, capture the smoke plume. The red star indicates the position of the Potenza lidar station, (center) TROPOMI ALH and (right) UVAI retrievals at 340/380nm pair. Missing ALH pixels are flagged by a cloud or have negative AI values.

**Figure 10** (left) illustrates the temporal evolution of the observed aerosol plume by means of the time–height cross sections of the 1064 nm total attenuated backscatter coefficient for the time period from 12:00 UTC to 17:55 UTC on the 24[th] of October. As observed in these timeseries obtained from PollyNet (https://polly.tropos.de/, last access: 01 May 2022), a significant particle load is detected after 11:00UTC, at 10-12km approximately. The red dashed box indicates the temporal averaging of these lidar signals close to the S5P overpass time. The Portable Aerosol and Cloud Lidar (PAOLI) installed at the Évora Atmospheric Sciences Observatory (EVASO) (38.57°N, 7.91°W; 293 m a.s.l.) is a multiwavelength Raman lidar of the type Polly[XT] (Baars et al. 2016; Salgueiro et al., 2021) part of the European Aerosol Research Lidar Network (EARLINET). Backward trajectories shown in **Figure 10** (right), generated with the HYSPLIT model, were used to determine the origin of the air masses carrying aerosol plumes arriving at the Évora site at the relevant heights (7500, 9500 and 11500m). They confirmed that the relevant air masses came from areas over North American, Californian forest fires detected by VIIRS.

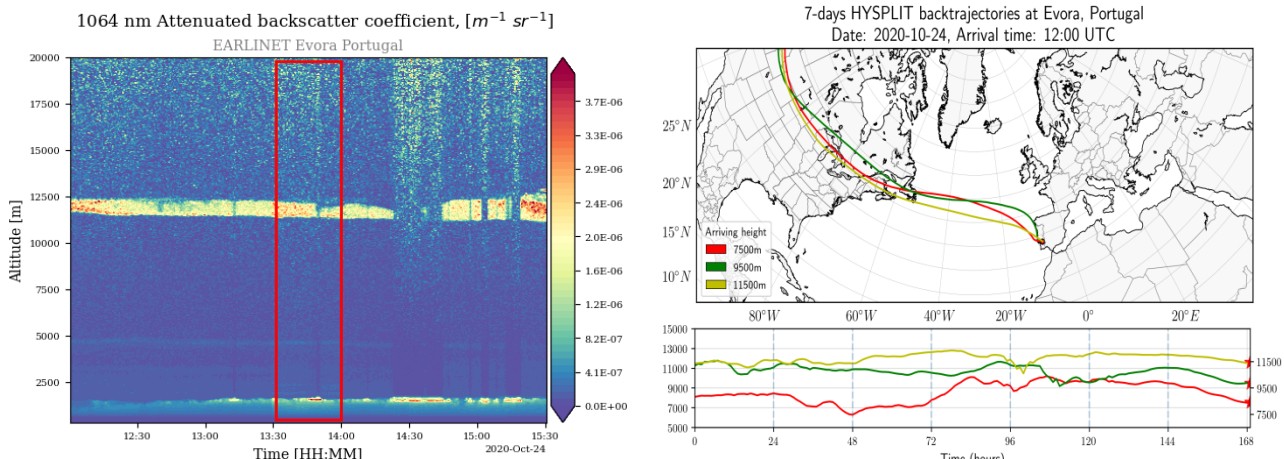

**Figure 10.** (Left) Temporal evolution of the total attenuated backscatter signal from the PAOLI PollyXT lidar system at 1064nm showing the detection of the smoke cloud. (Right) 7-day HYSPLIT back-trajectories arriving at Évora, Portugal on 24 October 2020 at 12:00 UTC.

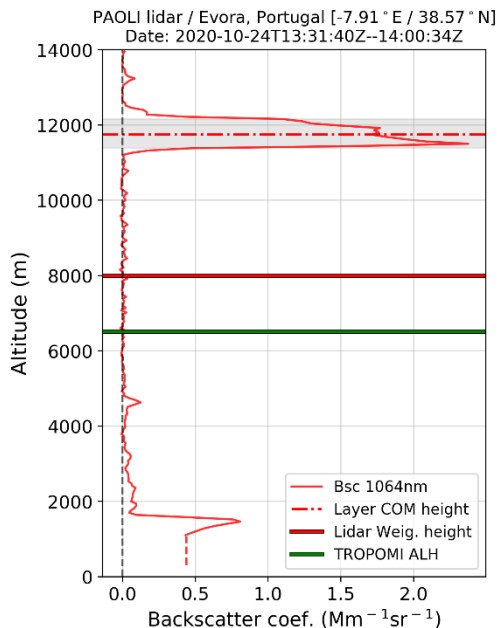

**Figure 11**. As per Figure 5 for the PAOLI lidar observation on 24 October 2020.

In **Figure 11** the retrieved vertical profile of the observations with the PAOLI PollyXT lidar is presented. The temporally closest backscatter profile is used to extract the $ALH_{bsc}$ and compare against the TROPOMI ALH retrievals. The average backscatter profile at 1064 nm, for the time period from 13:30 to 14:00 UTC on 24 October 2020 is shown. The TROPOMI observations report an average layer at 6500m, while the calculated $ALH_{bsc}$ (following Eq. 1) from the lidar profile places it at 7980m. As above, we also applied the WCT method on the backscatter vertical profile, in order to extract the aerosol boundaries of the detected aerosol layers in automatic way. The WCT technique also reveals a clear single layer between 11000 and 12500m with maximum backscatter value ~1.8 $Mm^{-1}sr^{-1}$ and center of layer mass at 11800m. For this case, we note the large discrepancy of both lidar layer identification techniques and the TROPOMI ALH whose reasons warrant further investigation in the future.

- **Smoke case over central Mediterranean: 26 October 2020, Potenza (Italy)**

On October 26th, the same smoke plume spread towards the central Mediterranean due to the easterly prevailing winds across Italy and Greece. In this sub-section, we present a case study within this smoke episode, over the Potenza lidar station in Italy for the 26th of October 2020. A significant aerosol load is observed mainly over the south of Italy. The true colour image (**Figure 12**, left) captured by VIIRS aboard Suomi-NPP, provides the context for the retrievals shown next. The location of the smoke plume is clearly seen in the TROPOMI ALH and UVAI images (**Figure 12,** center & right) during the Sentinel-5P overpass between 11:20-12:20 UTC. The TROPOMI UVAI shows a wide range of values with several patches with no retrievals due to the presence of clouds. This case is also emphasizing the issue related to the limitation of

satellite measurements over land areas where the effect of the surface reflectance is dominant. The contrast observed between land and sea regarding the retrieval of the ALH product and the surface albedo values is obvious as can be seen from the color scale in ALH retrievals. The ALH retrievals are very clearly biased over land, wherethe high surface albedo biases ALH low.

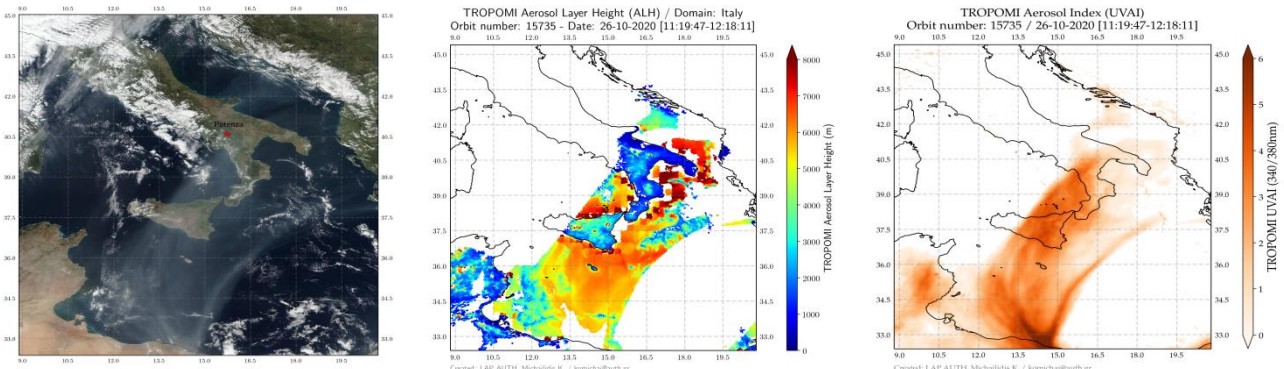

**Figure 12**. (Left)VIIRS Suomi-NPP True color image of 26 October 2020 over central Mediterranean, capture the smoke plume. The red star indicates the position of the Potenza lidar station. (Center) TROPOMI ALH and (right) UVAI retrievals. Missing ALH pixels are flagged by a cloud or have negative AI values.

The event is extensively recorded on October, 26[th], 2020 at the Potenza lidar station. MUSA is the lidar system (Madonna et al., 2011) deployed at CNR-IMAA Atmospheric Observatory (CIAO) in Potenza (40.60°N, 15.72°E, 760 m a.s.l.). **Figure 13** (left) shows the total attenuated backscatter time-series at 1064 nm measured by the MUSA system during the smoke event. The lidar observations started on 26 October 2020 at 10:00 UTC and lasted almost continuously until 13:30 UTC. The red dashed box indicates the temporal averaging of the lidar signals (10:00–11:30 UTC) close to the TROPOMI/S5P overpass time. Multilayer structures were found, and smoke particles appeared in the free troposphere, between 6000 and 11000 km above sea level (a.s.l.). The intense part of the smoke plume is located about 300 km south of the Potenza EARLINET station where the atmospheric conditions are different with different TROPOMI retrievals from above the station. The temporal evolution of 7-day backward trajectories, for this day (arrival heights: 7500 (red), 9500 (green) and 11500 m (yellow) is illustrated in **Figure 13** (right). As can be seen, the air masses which arrived over Potenza station seem to originate from N. America, following an almost straight route path towards Italy.

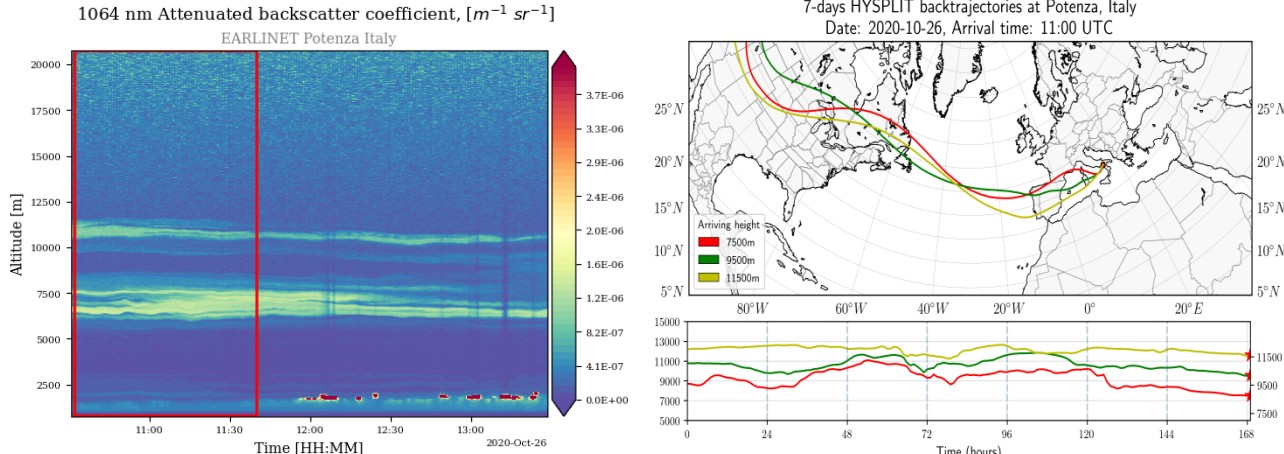

**Figure 13**. (Left) Temporal evolution of the total attenuated backscatter signal from the MUSA lidar system at 1064nm showing the detection of the smoke cloud. (Right) 7-day HYSPLIT back-trajectories arriving at Potenza, Italy on 26 October 2020 at 11:00 UTC.

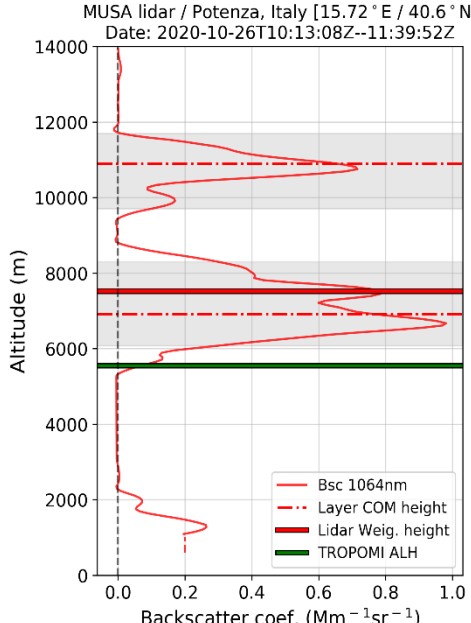

**Figure 14.** Same as Figure 5 for the MUSA lidar observations on 26 October 2020.

In **Figure 14** the retrieved vertical profile of the observations with MUSA lidar is presented. The closest in time backscatter profile is used in order to extract the $ALH_{bsc}$ and compare against TROPOMI ALH retrievals. The averaged backscatter profile at 1064 nm, for the time period from 10:15 to 11:40 UTC on 24 October 2020 is shown. Two optical elevated thick layers with a thickness of ~2km were detected with TROPOMI detecting this layer at 5650m while the calculated $ALH_{bsc}$, from applying Eq. 1 on the lidar profile, places it at 7800m. For this case which shows two well-developed layers, the retrieved geometric properties by applying the WCT technique place the first detected layer at 7150m and the second at 11070m, also with thickness of ~2km. As previously, we also note here a discrepancy between both lidar aerosol heights and the satellite estimation of the aerosol height. Clearly, the AER_LH is placed much lower than both calculated altitudes of the lidar profile.

This case of very high altitude smoke from intense biomass burning in North America in 2020 shows a notable difference with lidar measurements, revealing a source of limitations of the current operational S5P L2_AER_LH product. The current implementation of the algorithm is based on a neural network forward model and an optimal estimation scheme in the retrieval for spectral fitting with various aerosol layer pressures and aerosol optical thicknesses in the $O_2$A-band. The limitation exists due to the fact that the algorithm has not been trained for very high altitudes and the versions of the algorithm used in our study do not take these altitudes into account, as shown for these two the cases of elevated smoke layers. Furthermore, scaling of the assumed pressure thickness must be considered for very high altitude (low pressures) aerosol plumes, observed mainly for biomass burning plumes and volcanic ash/sulphate aerosols. Currently, the ALH neural network is trained for ambient pressures between 1000 and 75 hPa, which is about 12 km altitude maximally, and plumes above these heights cannot be resolved.

# 4    Summary and Conclusions

The TROPOMI Aerosol Layer Height (ALH) is a new and unique product providing global observations of aerosol height. TROPOMI aerosol layer heights can add value to the modelling communities by improving air quality forecasting and radiative forcing studies. Aerosol plume heights from TROPOMI have the advantage of daily global coverage. This is the first work in which TROPOMI ALH retrievals are validated against EARLINET lidar observations and the results provide an early evaluation of their applicability for monitoring aerosol height levels in a large area such as Mediterranean and Europe. The choice of the EARLINET stations close to the sea, has been performed considering the limitation of the TROPOMI ALH algorithm over land surfaces in providing accurate retrievals. Using these sites presented an opportunity to study the effect of the surface albedo on the validation of ALH product performing separate analyses over land and sea. The data used as reference for the validation were not part of a specifically designed validation campaign, which explains the small number of collocated cases found. This paper presents a cross-comparison analysis between TROPOMI and EARLINET data and provides a simple and well-developed methodology for comparing these different datasets. Lidar instruments retrieve the vertical backscatter coefficient, which is not directly comparable to the TROPOMI ALH product. Thus, the weighted-backscatter height ($ALH_{bsc}$) has to be calculated from the available backscatter profiles in the frame of this study. Coincidences within a 150 km radius from the lidar station are used for direct observations, with a maximum of 4h time difference. All the input datasets considered in the study have been previously pre-processed at high resolution by using the EARLINET Single Calculus Chain (SCC). Overall, for 7 selected EARLINET stations across the Mediterranean, 63 coincident aerosol cases were found during the time period June 2018 -July 2022, for the comparison against satellite retrievals. The statistical results demonstrate the potential of the TROPOMI instrument to detect aerosol layers under cloud-free atmospheric conditions with significant aerosol load, such as dust and smoke plumes.

Despite the different measuring concept that two instruments used for retrievals (passive & active), a good agreement was found between TROPOMI retrievals and ground-based lidar measurements, demonstrating that TROPOMI shows a quite promising potential for the characterization of the aerosol vertical distributions on a global scale. By defining a backscatter-weighted aerosol height from EARLINET aerosol backscatter profile products ($ALH_{bsc}$), the quantitative validation at pixels over the selected EARLINET stations illustrates that TROPOMI ALH is consistent with $ALH_{bsc}$, with a high correlation coefficient R=0.82 (R=0.51) and a mean bias about -0.51±0.77 km (-2.27± 1.17 km) over ocean and land pixels respectively. It appears that aerosol layer altitudes retrieved from TROPOMI are systematically lower than altitudes from the lidar retrievals. The target requirement on accuracy and precision of retrieved aerosol layer height is 0.5 km or 50 hPa; the threshold requirement is 1 km or 100 hPa. Overall, our results show that the TROPOMI product complies, under conditions, with the S5P mission requirements while our findings are in a good agreement with other TROPOMI ALH validation studies. The outcomes of this study, including the findings of previous validation works are summarized in Table 6. Nanda et al. (2020) and Chen et al. (2021) also discuss the challenges associated with the validation of TROPOMI ALH. These challenges arise mainly from the large spatio-temporal variability of aerosols, the dependency of the products on different geophysical parameters (e.g. surface albedo) and different instrument sensitivities. The effect of the surface albedo has been investigated through the sensitivity tests presented in detail in previous studies (Sanders and de Haan, 2016; Ding et al., 2016; Dubuisson et al., 2009) covering a large number of surface albedo values and showed that including the surface albedo in the optimal estimation fit considerably improves the ALH in most cases.

Table 6. Summary of validation statistics for TROPOMI ALH using O2A algorithm.

| Reference | Validation Data | Results |
|---|---|---|
| Griffin et al., 2020 | S5P vs CALIOP | Mean bias of -2.12 km (thin smoke plumes)<br>Mean bias of -0.7 km (thick smoke plumes) |
| | S5P vs MISR | TROPOMI ALH is lower, by ~ 600 m |
| Nanda et al., 2020 | S5P vs CALIOP | Mean bias of -2.41 km over land / 1.03 km over ocean |
| This work | S5P vs EARLINET | Mean bias of -2.27 km over land / -0.51 km over ocean |

The Wavelet Covariance Transform method was also applied on the ground-based lidar profiles so as to quantify the effect of multi-layer structures on the comparisons between TROPOMI and lidar aerosol layer height. This method provides better insight as to the altitude ranges that the two instruments are sensitive to, however more case studies need to be analyzed in detail to draw conclusions concerning to which sensed layer the TROPOMI algorithm is more sensitive to.

This study highlights the importance of the synergistic use of active (ground-based lidars) and passive (satellite) observations and suggests a promising usage of TROPOMI ALH for understanding the details of the presence and transport of aerosol layers. The results presented here encourage the operational usage of the presented methodology approach in validation processes for satellite aerosol height products using lidar data from EARLINET. The increased availability of advanced and high quality-assured profiling data from EARLINET lidars will form a scientific background to improve performance of passive satellite sensors and lead to a better understanding of the role of the aerosol height on air-quality and the climate. The inclusion of more stations from continental Europe will improve the significance of the results and will allow to study the impact of different aerosol types on the comparisons. In addition, it will make feasible to examine possible geographical dependencies. Lidar measurements within the EARLINET network are continuously performed and can be used in the coming years for the validation of the new satellite generations, such as the Copernicus Sentinel 4 and Sentinel 5.

**Data availability.** The EARLINET data used in this study are available from the authors and upon registration from the EARLINET web page at https://data.earlinet.org/earlinet/login.zul (Pappalardo et al., 2014). TROPOMI AEH data are available from the S5P Pre-Operations Data Hub at https://s5phub.copernicus.eu/dhus, (last access: 01 May 2022).VIIRS/Suomi satellite data are available online at https://wvs.earthdata.nasa.gov, (last access: 01 May 2022). HYSPLIT data as described by Stein et al. (2015) can be found at https://www.ready.noaa.gov/HYSPLIT.php (last access: 01 May 2022)

**Acknowledgments.** We are grateful to all Co-PIs of the EARLINET sites used in this study for maintaining their instruments and providing their data to the community. This research was supported by data and services obtained from the PANhellenic Geophysical Observatory of Antikythera (PANGEA) of the National Observatory of Athens (NOA), Greece. We thank the PollyNet group, and especially Ronny Engelmann and Holger Baars, for their support during the development and operation of the PollyXT lidar of NOA. NOA team acknowledges the support of Stavros Niarchos Foundation (SNF). R-E M and AN acknowledge the ERATOSTHENES Centre of Excellence and the "EXCELSIOR" H2020 Widespread Teaming project that has received funding from the European Union's Horizon 2020 Research and Innovation programme under grant agreement no. 857510 and from the Government of the Republic of Cyprus through the Directorate General for the European Programmes, Coordination and Development. DB, MJC and VS are co-funded by national Portuguese funds through FCT - Fundação para a Ciência e Tecnologia, I.P., in the framework of the ICT project with the references UIDB/04683/2020 and UIDP/04683/2020, as well as through TOMAQAPA (PTDC/CTAMET/29678/2017) and CILIFO (0753_CILIFO_5_E) projects.

**Author contributions.** **KM** carried out the processing of satellite and lidar measurements, prepared the figures of the manuscript and wrote the original draft of the manuscript with contributions from all co-authors. **MEK** and **DB** were responsible for the methodology and conceptualization of the paper. **LM, NP, EM, IT, AG** perform the lidar measurements ensured the provision of the QA EARLINET data. **PV** and **MdG** worked on the development of the TROPOMI AER_LH product and were responsible for providing satellite data, detailed description of the product. **KM** contributed to the development of the automatic algorithm for the aerosol layer detection using lidar data. **AP, R-EM, MM, LAA, DB, MJC, VS, SR, SRP, MRP** reviewed the case studies of the selected EARLINET stations, as presented in the paper. All authors participated in scientific discussions on this study, reviewed &edited the manuscript during its preparation phase.

**Financial support.** This research has been supported by the "Panhellenic Infrastructure for Atmospheric Composition and Climate Change" project (grant no. MIS 5021516) which is implemented under the Action "Reinforcement of the Research and Innovation infrastructure" and co-financed by Greece and the European Union (European Regional Development Fund). The writing and editing of this paper was carried out as part of the ESA-funded Quality Assurance for Earth Observation (IDEAS-QA4EO) framework contract. The authors also acknowledge the financial support of the European Space Agency "Preparation and Operations of the Mission Performance Centre (MPC) for the Copernicus Sentinel-5 Precursor Satellite". The Limassol, Cyprus, observations have been supported by the SIROCCO project (grant no. EXCELLENCE/1216/0217) and AQ-SERVE project (INTEGRATED/0916/0016) co-funded by the Republic of Cyprus and the structural funds of the European Union for Cyprus through the Research and Innovation Foundation. The PollyXT-CYP lidar funded by the Federal Ministry of Education and Research (BMBF) via the PoLiCyTa project and its operation by the EU H2020 EXCELSIOR project. EM and VA were supported by the European Research Council (ERC) under Community's Horizon 2020 research and innovation framework programme-ERC grant agreement no. 725698 (D-TECT)

**Competing interests.** The authors declare that they have no conflict of interest.

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
