# Peer review of "Validation of the TROPOMI/S5P Aerosol Layer Height using EARLINET lidars"

_Atmospheric Chemistry and Physics, 2022_

## Author Response (AR1)

**Comment on acp-2022-412**

Anonymous Referee #2

Referee comment on "Validation of the TROPOMI/S5P Aerosol Layer Height using EARLINET lidars" by Konstantinos Michailidis et al., Atmos. Chem. Phys. Discuss., https://doi.org/10.5194/acp-2022-412-RC1, 2022

The authors present a quantitative evaluation of the accuracy of the aerosol layer height (ALH) product derived from the satellite-based Sentinel 5P-TROPOspheric Monitoring Instrument (TROPOMI-SP5) using ground-based lidar observations submitted to the European Aerosol Research Lidar Network (EARLINET) database. The study is focused on the Mediterranean Basin in which observations from 7 EARLINET stations are selected, taking into consideration their proximity to the sea and the presence of absorbing aerosols. Within a 3-year time frame the authors have found 34 suitable cases for the comparison which shows the challenge in satellite validation attempts but also marks the importance of networks such that of the EARLINET. Given the importance of the ALH information in radiative forcing calculations, UV aerosol index, aviation safety etc. the study has scientific interest and therefore it is worth publishing. The work is overall sound, and I have only a few points to raise.

We would like to thank the reviewer for his/her fruitful comments that led to the improvement of the manuscript. In the following, answers to comments are reported just below each related comment. When needed, the part of the manuscript we modified or added to the old version, is reported.

General changes to the manuscript:

- In the revised version, new collocated cases have been identified and added in the analysis. We have added twenty-nine (29) more validation cases providing additional statistical significance in our validation results. Now the final collocated cases are 63, extending the time period to July 2022.

- In the revised manuscript, we separated the comparison between S5P and EARLINET for satellite pixels over sea and land.

The manuscript is well structured, but I miss a thorough discussion on the findings. Why the comparison is worse over the land/ocean dataset compared to the ocean? Mention previous studies which have already found this known feature of TROPOMI retrievals and make comprehensive conclusions.

The main reason for the underestimation of layer height by the TROPOMI sensor over land is the surface reflectivity assumed in the forward model primarily due to the high surface reflectance in O2A-band that is not favorable for aerosol retrievals. In the revised manuscript we improved the discussion of the finding taking into account the reviewer's comments/suggestions. Concerning the poor performance of TROPMI over land we added in the discussion the following sentence in **Section 2**: "The main reason for the strong underestimation of the aerosol layer height retrieved by the current algorithm from TROPOMI over land is the surface reflectivity climatology used in the forward model, leading in biased or non-convergent retrievals over land. Sensitivity studies showed that the observed large bias over land is reduced when fitting of the surface albedo as estimated from TROPOMI itself was included in the retrieval procedure. This will be further investigated in the near future and is

intended to be implemented in future versions of the ALH L2 product"

Concerning previous studies, we added a new table in the end of the discussion section (Sect. 4) summarizing the outcomes of this study, including the findings of other works (Griffin et al., 2020; Nanda et al, 2020). The present analysis provides important additions to the existing validation studies that have been performed so far for the TROPOMI S5P ALH product, which were based only satellite-to-satellite comparison (e.g. CALIOP and MISR) and confirms in a consistent way the effect of surface albedo in the retrieved ALH. Moreover, the use of high-resolution ground-based lidar data makes feasible a better characterization of the biases found

What is missing from this comparison which would be beneficial in the community?

The distribution of the EARLINET stations allows us to study the temporal, regional and continental-scale representativeness of the observations and to compare these findings with the results of spaceborne passive instruments. However, the limitations of the current version of ALH algorithm over land does not allow a full exploitation of its potential. The inclusion of more stations from continental Europe will improve the significance of the results and will allow to study the impact of different aerosol types (e.g urban, rural, etc) on the comparisons. In addition, it will make feasible to examine possible geographical dependencies.

The Section 4 in the manuscript has been modified highlighting the main points of the above discussion. We think that pinpoints the importance of this work.

In this study, the geometrical features from ground-based lidars were compared against the ALH product from TROPOMI. I was wondering since most of the ground-based lidars used in this study have a lower detection limit at around 700-800m, if not higher, how was this taken into consideration when calculating the ALH? How did the authors tackle the overlap issue in the ground-based lidar observations and what is the error from ground-based lidar overlap limitation to the calculated ALH? For example, in the smoke episode the lidar signal is cut below 1km (Figure 11 and Figure 14). I assume that the bias is not big (equation 1) but given that the attempt is to validate a satellite product this effect should be discussed.

The reviewer raises a crucial issue regarding how the overlap altitude can affect the lidar-based ALH estimates. In the initial submission we did not consider the effect of the overlap in the estimation of the ALH from a lidar measurement.

To overcome this issue, in the revised version, we rely on certain assumptions. To calculate the $ALH_{bsc}$ from the lidar backscatter profiles using equation 1 (Sect. 2.3), for the height range between the surface and the full overlap height we assumed a constant backscatter coefficient (height-independent) equal to the one measured at the full overlap height. This is general acceptable since the Planetary Boundary Layer (PBL) is characterized as well-mixed aerosol conditions (Siomos et al., 2018). This assumption obviously affects the calculation of the lidar aerosol height ($ALH_{bsc}$) compared to the ones shown in the initial submission since it also considers the contribution of the aerosol load in the lowermost part of the atmosphere. The $ALH_{bsc}$ estimates, when considering this part of the profile, are therefore smaller and the bias with TROPOMI is reduced.

In order to quantify the uncertainty of the $ALH_{bsc}$ estimates due to assumption we made for tackling the incomplete overlap issue we performed a number of sensitivity tests. An example case over Thessaloniki on 15 June 2022 is presented in the **Figure RC1-1** which demonstrate the effect of the overlap altitude on the lidar ALH calculation for different scenarios. Dashed colored lines correspond to the different indicative assumptions for the backscatter coefficient profile below the overlap height. The horizontal-colored lines indicate the corresponding lidar

weighted height. The scenario with a negative slope of backscatter in lower part is indicated in green, the case for vertical extension (zero signal slope) in red, and the third with a positive slope in blue. After applying the above sensitivity tests to all lidar measurements (N=63) used in the study, we can concluded that the effect of the different assumptions shown on the calculation of $ALH_{bsc}$ is of the order of 100 - 400m, depending on the technical characteristics of each lidar system.

It should be also noted that in the calculations described above, the altitude of the EARLINET stations is considered for the calculation of the $ALH_{bsc}$. Most stations are located at low altitude in coastal areas, so it does not play a significant role, in contrast to stations located at an altitude > 600m such as Granada and Potenza, where the effect is significant.

In the revised manuscript, a new paragraph has been added to present the main points of the discussion above.

[Figure]

**Figure RC1-1**. Sensitivity test to define the effect of overlap altitude on $ALH_{bsc}$ calculation under different scenarios (Thessaloniki 15 Oct 2022)- Dashed lines correspond to the different overlap assumption.

During daytime, the Klett method was used for the retrieval of the particle optical properties. I was wondering how the selection of a single lidar ratio (LR) for the whole profile can skew the ALH calculation in the presence of multiple aerosol layers in which the aerosol type is not the same since it affects the particle backscatter coefficient value and therefore the ALH calculation? How many of the 34 cases were Raman cases? Was there any difference in the bias between the Raman cases and the Klett cases?

We use the lidar backscatter coefficient profiles at 1064 nm (or 532 nm), analyzed by the Single Calculus Chain (SCC; https://scc.imaa.cnr.it/) algorithm (D'Amico et al., 2016) for quality-assured measurements. Raman measurements are not used in the study. The Klett-Fernald-Sasano (KFS) inversion is applied (Klett,1981; Fernald,1984; Sasano and Nakane,1984) to retrieve the height-resolved aerosol backscatter coefficients with selection of constant lidar ratios in most of cases, based on the climatology obtained from each station. Following the reviewer's suggestion, we studied the effect on the $ALH_{bsc}$ estimates based on lidar backscatter profile retrievals for different Lidar Ratio (LR) values (20-60sr). We assumed: (a) constant values of the lidar ratio with height and (b) LR height dependent profile. The results show that in both cases the effect of the different lidar ratio values on the weighted height calculation (ALHbsc) is small, lower than 40m.

A relevant phrase has been added in the manuscript.

**Technical corrections:**

In general, the writing of a scientific article should be impersonal, therefore, I would recommend rechecking these places in the manuscript were the word 'we' has been used. To this direction, there are a few typos and misuse of English language in several places in the manuscript. A careful review is required.

Rephrased

L38: Repentance of the text 'over the Mediterranean basin'. Please, correct.

Corrected.

L43: '…illustrates that TROPOMI ALH is consistent with EARLINET'. A satellite product (ALH) cannot be consistent with a network (EARLINET). Rephrase the sentence.

Rephrased.

L54: 'Aerosol properties are one…' à 'Aerosol properties present one…'

Rephrased.

L77: Nanda et al., (2020) presented a comparison between TROPOMI ALH and CALIOP observations therefore not relevant in the context of this sentence.

We added this reference here as, to our knowledge, it is the most comprehensive paper currently available that describes the TROPOMI ALH. We did not add it as a validation reference.

L80: The EARLINET acronym is already defined in P2/L63. Similarly, in P4/L33.

Corrected.

L139. 'EARLINET measurements…' - > '. Observations submitted to EARLINET database follow….'

Corrected.

L149: 'On the other, during nighttime' -> 'During nighttime,…'

Entire phrase was deleted in the revised manuscript.

L166-L175: I suggest removing this paragraph as its context is not relevant to EARLINET. Section 2.2 or/and the discussion are more relevant candidates.

The paragraph has been removed following the reviewer suggestion. A new re-formulated paragraph added in the discussion section and in Section 2.2

L203: Please, provide the acronym for TOA.

Added.

L208: Provide a reference for OMI/Aura and their corresponding acronyms.

Acronym and reference provided.

L220: 'To construct…TROPOMI observations. Please, rephrase the sentence.

Rephrased.

L251: 'In addition, $ALH_{ext.......}$'. Do the authors refer to the weighted-extinction height? Please, specify.

The authors refer to the "weighted-backscatter height $ALH_{bsc}$ ". Text corrected.

L270: The acronym $Z_{COM}$ is already defined in L246. Please, go through the manuscript and carefully correct the usage of the acronyms. Define an acronym once and then use in the rest of the manuscript. Also give the acronyms that are missing e.g TOA, OMI etc.

Zcom has indeed been previously defined, but in these lines all the different Zs are given explicitly. So as to help the reader follow our work without unnecessary back-and-forth in the text, we opted to also give Zcom a second time.

TOA and OMI are now provided.

L278: 'In the case where more than one layers with a significant contribution to the optical thickness of the profile, an average value.…retrievals' à 'In case more than one layers with significant contribution to the optical thickness of the profile are present,…..'

Rephrased.

L286: 'two selected' à Two or three? In some places it is mentioned 2 in some others 3. I assume three is the correct answer.

It is indeed three, one dust and two smoke episodes

L388: Add UTC next to the time and the corresponding Fig. 4a.

UTC added next to the time.

L559: 'All the input datasets considered in the study have been previously pre-processed at high resolution'. What is high resolution referring to?

The EARLINET lidar profiles (e.g. backscatter profiles in our study) are provided with a height resolution of a few tens of meters (7.5 to 60m) and a temporal resolution of a few minutes.

**Review of manuscript 'Validation of the TROPOMI/S5P Aerosol Layer Height using EARLINET lidars', (acp-2022-412) by Michailidis et al. (2022)**

**Summary**

This manuscript presents results of comparisons of aerosol layer height (ALH) derived from Oxygen-A-band ($O_2A$) observations by the Sentinel5-Precursor Tropospheric Monitoring Instrument (TROPOMI) to ALH's inferred from the vertical distribution of aerosols by European Aerosol Research Lidar Network (EARLINET). Lidar observations at seven EARLINET stations along the Northern Mediterranean coastline yielded 34 Lidar-TROPOMI coincidences. A coincidence defined as the averaged TROPOMI ALH inside150 km radius circles centered at the EARLINET site within an 8-hr window ($\pm$ 4hr). The lidar inferred ALH is calculated as the 1064 nm backscatter-weighed aerosol altitude.

We would like to thank the reviewer for his/her fruitful comments that led to the improvement of the manuscript. In the following, answers to comments are reported just below each related comment. When needed, the part of the manuscript we modified or added to the old version, is reported.

General changes to the manuscript:

- In the revised version, new collocated cases have been identified and added in the analysis. We have added twenty-nine (29) more validation cases providing additional statistical significance in our validation results. Now the final collocated cases are 63, extending the time period to July 2022.

- In the revised manuscript, we separated the comparison between S5P and EARLINET for satellite pixels over sea and land.

**General Comments**

Although the authors have generally carried out a carefully planned validation analysis of TROPOMI ALH, a few details need additional explanation before the article is acceptable for publication.

-The 8-hour temporal window, dictated by the need of getting enough information for the analysis, is probably too long to capture the variability of the dust plumes' structure. If the frequency of EARLINET observations allow it, I suggest adding a figure documenting the typical variability of aerosol load vertical structure during the passage of a dust storm at a representative site. This analysis could provide important information to characterize the uncertainty associated with the adopted ALH definition.

We present in detail in our response to Reviewer 3 (comment 20) the time difference between the lidar measurement and the TROPOMI overpass. The majority of the measurements used are within +/- 2 hours and only two of them extend to +/- 4 hours. However, these cases correspond to persistent dust episodes and were considered useful for the comparisons. Following the

reviewer's suggestion, we present in detail such an intense dust episode originating from Saudi Arabia, on the 24 April 2022 over Eastern Mediterranean, closely to Cyprus. The temporal evolution of the range-corrected signal (RCS) and volume linear depolarization ratio (VLDR) over Limassol and illustrated in the **figure RC2-1**. The plots confirm that presence of a persistent dust layer located in the height range between 2 to 4km, above sea level, for many hours before and after the S5P overpass. The averaged backscatter profiles at 1064 nm (m$^{-1}$·sr$^{-1}$), calculated with an integration time of 60 min, for the time windows from 07:00 to 08:00, 09:00 to 10:00 and 12:00 to 13:00 UTC (see the yellow vertical dashed boxes), are presented on the upper part, revealing the differences in the vertical structure of the atmosphere between the different time windows. As it is evident from this figure the ALH$_{bsc}$ ranged from 3.1 – 3.3km.

[Figure]

**Figure RC2-1.** A dust storm over Eastern Mediterranean captured by the Polly$^{XT}$ lidar system operated in Limassol, Cyprus on April 24, 2022 (ADD better description). VIIRS/Suomi NPP True color image on this day also illustrated, generated from NASA Worldview Snapshot (https://wvs.earthdata.nasa.gov/)

-The authors mentioned having considered the UVAI as an indicator of the presence of absorbing aerosols. Please explain the manner the UVAI was used considering the UVAI dependence on height, in addition to the known dependence on AOD and composition. Just having a positive UVAI value is not enough to assume aerosol presence because there are other non-aerosol related sources of positive UVAI values.

An in detail explanation at this point is included in the revised manuscript.

- Given the limited set of coincidences as well as the localized nature of the analysis, I disagree with the authors' over-optimistic conclusion that the TROPOMIALH product meets the expected 1 km threshold requirement of either accuracy or precision. A previously published evaluation analysis (Nanda et al, 2020) demonstrated that the TROPOMI ALH product is systematically lower than CALIOP over both land and oceans.

Taking into account the reviewer's comment, in the revised version of the manuscript we reformulate the text accordingly so that it clearly reflects our claims. The reviewer can also refer to our response to comment of reviewer RC1 & RC3. In the revised version the estimated $ALH_{bsc}$ are smaller and reduce the bias to $0.51\pm0.77$ km as seen in the revised Figure RC3-2 (Rev. comment

-There was not mention of the possible role of calibration. It is well known that the calibration of the sensor has been drifting. The authors should address this issue and explain how calibration effects could (or could not) explain the observed level of disagreement in retrieved ALH.

We think that calibration issues are clearly beyond the scope of this manuscript. The technical documentation of the TROPOMI ALH provides error analyses and among many other issues, it describes the effects of instrument errors on the retrieval. The L1B radiances and irradiances are calibrated but will be corrected for degradation effects, in the foreseen v3.0. For this new version of the data an updated validation will be performed, which will allow such an assessment. More details about the calibration aspect can be found in Section 5 on the ATBD; http://www.tropomi.eu/sites/default/files/files/publicSentinel-5P-TROPOMI-ATBD-Aerosol-Height.pdf.

**Specific Comments**

Line 46. The statement 'This work confirms that the TROPOMI ALH product is within the required threshold accuracy and precision requirements of 1 km' should be removed. The temporarily and spatially limited analysis presented in this work does not provide any basis for such a general and over-reaching.

This comment has been replied to previously in this review.

Line 75. Add TEMPO to the list of upcoming AQ satellites

TEMPO is added in the text. Also, the reference below has been added in the manuscript.

Zoogman, P., Liu, X., Suleiman, R. M., Pennington, W. F., Flittner, D. E., Al-Saadi, J. A., Hilton, B. B., Nicks, D. K., Newchurch, M. J., Carr, J. L., Janz, S. J., Andraschko, M. R., Arola, A., Baker, B. D., Canova, B. P., Chan Miller, C., Cohen, R. C., Davis, J. E., Dussault, M. E., Edwards, D. P., Fishman, J., Ghulam, A., González Abad, G., Grutter, M., Herman, J. R., Houck, J., Jacob, D. J., Joiner, J., Kerridge, B. J., Kim, J., Krotkov, N. A., Lamsal, L., Li, C., Lindfors, A., Martin, R. V., McElroy, C. T., McLinden, C., Natraj, V., Neil, D. O., Nowlan, C. R., O'Sullivan, E. J., Palmer, P. I., Pierce, R. B., Pippin, M. R., Saiz-Lopez, A., Spurr, R. J. D., Szykman, J. J., Torres, O., Veefkind, J. P., Veihelmann, B., Wang, H., Wang, J., and Chance, K.: Tropospheric emissions: Monitoring of pollution (TEMPO), J. Quant. Spectrosc. Ra., 186, 17–39, https://doi.org/10.1016/j.jqsrt.2016.05.008, 2017.

Line 87 TROPOMI aerosol *height* products

Added

Line 210 Over the oceans positive UVAI also result from non-aerosol sources such as sunglint

and ocean color effects. Negative UVAI can also result from optically thin clouds and aerosols over both land and oceans. Ocean color effects associated with chlorophyll absorption yield negative values over the oceans. Surface spectral dependence over arid and semi-arid regions also generate non-aerosol related UVAI signal. Include original references [Herman et al., 1997; Torres et al 1998] and recent references documenting improvements in the treatment of water clouds in UVAI calculation [Torres et al, 2018]

In the revised manuscript a relevant phrase was added at this point.

Line 244 Elaborate on the cloud screening applied to TROPOMI observations for the comparison to EARLINET observations.

The ALH is very sensitive to cloud contamination. However, aerosols and clouds can be difficult to distinguish. We follow the PRF recommendations to identify possible cloud-contaminated pixels for aerosol retrievals. Cloud masks are available from VIIRS and FRESCO and is strongly recommended to filter for residual clouds. For the current algorithm versions, an operational cloud mask for the off-line processing mode is based on observations by VIIRS aboard Suomi-NPP. S5P flies in formation with Suomi-NPP and observes within approximately 5 min from Suomi-NPP's overpass. These and other sources of uncertainties are indicated with the "qa_value". Use of pixels with a "qa_value" below 0.5 is not recommended. Moreover Cloud flags are available and are strongly recommended to filter for residual clouds. For cloud filtering, the "cloud_warning" flag is the preferred flag for removing possibly cloudy pixels. The flag of "cirrus_reflecatnce_viirs_filter" for residual cirrus clouds is also used. In our case we apply all the above flags to the satellite retrievals.

An extended list of applying cloud flags to filter out possibly cloudy pixels is also provided in detail in Nanda et al., 2020 (Table 1). A full description of the pixel selection scheme is provided in product ATBD and PRF.

Line 245 The term *real* aerosol height is not appropriate in this context. Perhaps *effective* is a better choice.

Altered.

Line 246 Use 'backscatter-weighted' instead.

Altered

Line 445 Add the Torres et al [2018] reference that specifically addresses the treatment of clouds in the UVAI parameterization.

A relevant phrase was added at this point. In addition, the suggested reference has been added in the revised manuscript.

Torres, O., Bhartia, P. K., Jethva, H., and Ahn, C.: Impact of the ozone monitoring instrument row anomaly on the long-term record of aerosol products, Atmos. Meas. Tech., 11, 2701–2715, https://doi.org/10.5194/amt-11-2701-2018, 2018.

Line 538 It is reasonable to assume that the TROPOMI AER_LH algorithm development team are familiar with the thermodynamic nature of the Earth's atmosphere. So, either remove the sentence '…The AER_LH algorithm was not created to retrieve AER_LH at such low air pressures' or add a specific reference in support of that statement.

Obviously, the statement here needs rephrasing to avoid any misunderstanding. and in addition relevant references have been added in support of the statement, already mentioned in the article. The main point here is (see details below) that the limitation for low pressures exists not because of the thermodynamics but due to the fact that the current algorithm has not been properly trained for very high altitudes (i.e.. low pressures). A relevant explanation is added in the revised manuscript.

The current implementation of the algorithm is based on a neural network forward model and an optimal estimation scheme in the retrieval for spectral fitting with various aerosol layer pressures and aerosol optical thicknesses in the oxygen A-band. The ALH is reported in both altitude and pressure. The limitation for low pressures (i.e. high altitudes) exists because the algorithm has not been trained for very high altitudes. This is perceived by the cases of elevated layers (e.g. smoke, volcanic ash, sulphates) detected by TROPOMI. Currently, the ALH neural network is trained for ambient pressures between 1000 and 75 hPa, and plumes above these heights cannot be resolved.

The discussion above is included in the TROPOMI ATBD. The relevant reference is provided into the manuscript.

Line 542 The analysis presented in this paper shows a real limitation of the TROPOMI AER_LH algorithm, not just a *possible* one.

Agreed, slightly rephrased.

Line 578 Accuracy and precision are two different concepts. Based on the presented results, this reviewer is not convinced that the 1 km threshold requirement of either accuracy or precision has been met. I believe additional analyses are needed.

This comment has been replied to previously in this review.

Line 579 The use of 'testify' is mainly a legal term. It is not appropriate in thisscientific context. It could be replaced with 'show'.

Altered.

**Comment on acp-2022-412**

Anonymous Referee #3

Referee comment on "Validation of the TROPOMI/S5P Aerosol Layer Height using EARLINET lidars" by Konstantinos Michailidis et al., Atmos. Chem. Phys. Discuss., https://doi.org/10.5194/acp-2022-412-RC3, 2022

The paper entitled "Validation of the TROPOMI/S5P Aerosol Layer Height using EARINET lidars" aims to investigate the ability of the Sentinel-5P TROPOspheric Monitoring Instrument (TROPOMI) to derive accurate geometrical features of aerosol layers, through implementation of ground-based observations from the European Aerosol Research Lidar Network (EARLINET). The article falls within the scope of "Atmospheric Chemistry and Physics". The authors utilized the database of EARLINET in order to present a statistical analysis of the ALH retrievals of TROPOMI. I would suggest publication, for the results are of interest for the scientific community implementing and working on S5P, however following major revisions.

We would like to thank the reviewer for his/her fruitful comments that led to the improvement of the manuscript. In the following, answers to comments are reported just below each related comment. When needed, the part of the manuscript we modified or added to the old version is reported.

General changes to the manuscript:

- In the revised version, new collocated cases have been identified and added in the analysis. We have added twenty-nine (29) more validation cases providing additional statistical significance in our validation results. Now the final collocated cases are 63, extending the time period to July 2022.

- In the revised manuscript, we separated the comparison between S5P and EARLINET for satellite pixels over sea and land.

**Comments:**

**1.** "The purpose of this study is to investigate the ability of the Sentinel-5P TROPOspheric Monitoring Instrument (TROPOMI) to derive accurate geometrical features of lofted aerosol layers on a continental scale".
Considering EARLINET, the network spans on a continental – European scale, with stations established and operated even beyond the continental boundaries in recent years. The study focuses on a part of the continent, the Mediterranean Sea region. Thus, I would suggest to the authors to replace the term continental whenever used in this concept.

We accept the suggestion of Reviewer. The text has been modified accordingly.

Moreover, it is not clear the reason behind focusing on the Mediterranean Sea region, for EARLINET offers a unique wealth of ground-based observations based on more than 30 established and regularly operated and maintained stations. In this way the correlative dataset of TROPOMI-EARLINET would provide a significantly more extended number of collocated cases, offering the protential to provide more robust results.

Please refer to our response to RC1 concerning the effect of surface albedo to the retrieved ALH from

TROPOMI. Initially we also considered continental stations in the comparisons, where no ocean pixels were available within a radius of 50-150km. After applying the recommended quality flags for the ALH product, and extremely small number of pixels were left to be compared with the ground-based estimates, which would not allow any meaningful comparisons. We chose to select stations relatively close to the sea in order to be able to compare separately the ocean and land pixel retrievals. In a future version of the algorithm we will also consider the rest of the stations, which would allow to provide a statistically significant analysis and findings (see the response). One other aspect was the availability of suitable EARLINET data during the period examined, which is affected by the fact that EARLINET measurements are systematically performed following a standard schedule (every Monday and Thursday) and not optimized for the validation of TROPOMI. There isn't a standard schedule between ESA and EARLINET for lidar measurements like in case of CALIPSO and Aeolus satellite missions.

The manuscript has been modified accordingly presenting the main points of the discussion above.

Even in terms of a study focusing on the "Mediterranean Sea" region, the Evora station is considered less a Mediterranean station than an Atlantic Ocean station, with a significant number of EARLINET stations falling within the Mediterranean and not included in the study.

We preferred to use the "Mediterranean" because this reflects the majority of the stations (6 out of 7). We consider that the Evora station fulfils the selection criteria presented before, the results are consistent with the other stations and strengthens our conclusions.

Please justify in the study the selection of the stations used and not used in more solid way, for the selection significantly affects the conclusions, due to the effect on the number of cases in the intercomparison.

See the response to the previous comment and also the comment 20.

**2.** "… key component in the validation of passive satellite aerosol product …". Please add "passive and active …".

Updated.

**3.** The introduction has to include quantitative outcomes of studies related to previous evaluation or validation of TROPOMI ALH. Moreover, the manuscript will benefit by a table in the end summarizing the outcomes of the study, including lines/rows with outcomes of previous studies (i.e. Nanda et al, 2020).

The authors agree with the reviewer that is would be beneficial to summarize the main results from the validation studies related to the S5P ALH product. For this reason, the following table was added in the Section 4 of the revised manuscript, including additionally the findings from the present work. The text in the revised manuscript has been modified accordingly.

**Table RC3-1.** Summary of validation results based on previous TROPOMI ALH using O2-A algorithm.

| Reference | Validation Data | Results |
|---|---|---|
| Griffin et al., 2020 | S5P vs CALIOP | Mean bias of -2.12 km (thin smoke plumes)
 Mean bias of -0.7 km (thick plumes) |
| | S5P vs MISR | TROPOMI ALH are lower, by approximately 600 m |
| Nanda et al., 2020 | S5P vs CALIOP | Mean bias of -2.41 km over land / -1.03 km over ocean |
| Michailidis et al. 2022 | S5P vs EARLINET | Mean bias of -2.27 km over land / -0.51 km over ocean |

**4.** prominent ->key.

Altered.

**5.** The manuscript could benefit from references to support the context, for very few are used. Though some examples are provided here of missing references, I would strongly suggest the authors to strengthen the manuscript with references, and go through the text carefully and enrich it.

… relatively short life … (add reference).

… variety of their natural and anthropogenic sources … (add reference).

… and their different formation mechanisms … (add reference).

… aerosols exhibit highly variable spatio-temporal distributions around the globe … (add reference).

… strongly affect both air quality … (add reference).

… the delicate balance in atmospheric chemistry … (add reference).

… is essential for understanding the impact of aerosols on the 60 climate
system … (add reference).

Both active and passive remote sensing methods have been developed, from both ground-based and space-borne systems, in order to estimate the aerosol layer height … (add reference).

... influenced by the Sahara Desert on the South and the highly populated and industrialized European countries to the North … (add reference).

... this relatively high aerosol load in the region can have strong effects on the regional radiative budget (add reference), climate (add reference), and ecosystems (add reference) ...

… with frequently observed events of mineral dust and smoke particle (add reference) … And more. Please go through the manuscript.

Since the article was enriched with text, it was carefully screened for such English mishaps. However, please recall that there is more than one way to express the same thing in the English language.

**6.** The English language of the manuscript is acceptable for publication; however, it is rather poorly used. The manuscript is characterized by a large number of non-formal, non-highly scientific approach of describing the core of the context of the manuscript. I suggest to the authors to improve the language of the manuscript. Only some examples are provided here, phrases that do not read well..

… to reduce uncertainties in our understanding …

… Space-based instruments are able to fill this gap …

… In order to trust and use the space-based products …

… Mediterranean Sea basin …

… The lidar technique is the most predominant tool …

… The large majority of the involved stations is based on multi-wavelength …

… and are equipped with depolarization channels EARLINET measurements follow absolute accuracy standards …

... ground-based lidar measurements from first need to be collected and collocated ...

… , 34 coincident cases were found, checked and flagged …

… The total available dataset is on the small side but suitable for the comparison study and general representativeness of the TROPOMI ALH product…

… ... Only a few data satellite points are available over the land and so a meaningful direct comparison over land only is not possible…

… Recall that ...

… This example amply demonstrates that when …

… The smoke arrived over the Iberian Peninsula (IP) in southwestern Europe on 24 October (Figure 6a), just in time for a regular overpass of the TROPOMI over Iberian Peninsula …

… the TROPOMI ALH whose reasons warrant further investigation in the future…Please go through the manuscript.

References added at the relevant pertinent parts of the article, as suggested.

**7.** Lines 77-79. Both validate and evaluate terms are used in a sentence. The two terms are different. The objective is to validate or evaluate TROPOMI ALH?

The sentence is confusing as it is written in the submitted text. We are clarifying better this point in the revised manuscript that the objective is to validate the ALH

**8.** Line 92: strategy -> methodology.

Altered.

**9.** Half of Section 1.1. has nothing to do with providing information on the Mediterranean Sea, for it discusses EARLINET. I would suggest moving the first part to the introduction and the second (EARLINET) to section 2.1.

The text has been slightly modified following the reviewer's suggestion.

**10.** Frequently the authors provide the extended acronym explaining the terms (or providing information on last access) two or more times in the manuscript, not necessarily during the first time when the terms are used. Please elaborate on the issue.

We found a number of these occurrences and corrected them in the updated article.

**11.** Line 138: … and Raman-shifted signals, and …

Please check for spaces between words and characters.

Checked.

**12.** Line 145: attenuated backscatter or backscatter coefficient?

Authors' refer to "backscatter coefficient"

**13. Major comment:**

Line 145: A major consideration is that though EARLINET is a high-level network and is considered as such the ground-truth for such studies, the present study has done a poor job in addressing sources of discrepancies in the comparison related to the characteristics of the reference dataset.

I would ask the authors to extensively discuss:

1) The errors of the EARLINET measurements in the study and how they are used and affect the comparison.

To ensure the homogeneity and consistency of the optical property profiles derived from each lidar system operating in each station, the Single Calculus Chain Algorithm (SCC; D' Amico et al., 2015;2016) was used. Herein elastically backscattered signals at 1064nm and 532nm, were used to calculate the $ALH_{bsc}$. The backscatter profiles are used from each station together with the associated error in the vertical profile. After applying the Monte Carlo error propagation using the backscatter profiles and the errors, for all the cases we found that the effect on the estimated $ALH_{bsc}$ is small of the order of some tens of meters, ranging between 10 - 60 m. A relevant sentence is added in the manuscript.

2) The strong effect of the overlap. In many cases, the stations are at relatively high altitude amsl, while an additional overlap may push the comparison even higher, or even over the PBL. Lower aerosol layers detected in the PBL/MBL are observed by S5P, not fully by EARLINET. Which is quantitatively the effect of overlap in the study?

The overlap altitude can influence the value of derived ALHs. The incomplete overlap between the laser beam and the receiver field of view significantly affects lidar observations of particle properties in the near-field range. Therefore we explore the influence to the results. This aspect has been discussed in detail in a previous comment. The effect of the overlap assumptions shown on the calculation of $ALH_{bsc}$ is of the order of 100 - 400m, depending on the technical characteristics of each lidar system. Please, see for details our response to RC1.

3) Topography plays a key role in satellite-ground based intercomparison of measurements/products. The different stations are characterized by different topographical features, affecting the homogeneity of aerosols in the comparison. A nice example is illustrated in Gkikas et al., 2022, "First assessment of Aeolus L2A particle backscatter coefficient retrievals in the Eastern Mediterranean" – Figure 1. In the present study, the same collocation criteria are used for stations of different characteristics.

We agree with the reviewer's statement that the topography can affect the satellite-ground based intercomparison of measurements/products. Over areas with a complex terrain, vertical inconsistencies between ground-based and satellite retrievals, due to possible orography induced disturbances in the aerosol layer height. It should be also noted that the use of a circular sampling area around the sites may not always be the optimal choice for the comparison since the Aerosol layer height variability is not necessarily symmetric due to the effects of local topography on the aerosol transport.

In order to study the effect of topography on the TROPOMI ALH retrievals the authors separated the EARLINET stations used in the study, into 2 separate clusters: Coastal (Limassol, Lecce, Antikythera, Athens) and Mountainous (Potenza, Granada, Evora). The locations and the characteristics of the stations are given in **Table 1** and **Figure 1** in the initial manuscript.

According to the findings, it is evident that the correlative measurements between the Mountainous EARLINET stations and the S5P overpasses are characterized by higher variability, more extreme differences, higher mean biases than in the Coastal cases. The complex topography, in terms of geographical characteristics, the horizontal distance between the TROPOMI retrieved pixels and the ground-based lidar sites possibly enhance the discrepancies resulting in higher differences between the EARLINET and TROPOMI values. The statistical metrics on the effect of topography are given in Tables RC3-2 and RC3-3 for ocean and land satellite retrievals separately. However, we should note that these finding are based only on few satellite retrievals and we will need more data to better justify this conclusion.

**Table RC3-2:** Clusters of EARLINET stations and TROPOMI comparison statistics for ocean pixels.

| TROPOMI pixels over ocean | | | | |
|---|---|---|---|---|
| **Clusters** | **Number of cases** | **R** | **MB [km]** | **Y=aX+b** |
| **Coastal stations:** AKY, ATZ, SAL, LIM, CYC | 46 | 0.78 | -0.47 | 0.69+0.40 |
| **Mountainous stations:** POT, GRA, EVO | 17 | 0.83 | -0.61 | 0.67x+0.55 |

**Table RC3-3:** Clusters of EARLINET stations and TROPOMI comparison statistics for land pixels.

| TROPOMI pixels over land | | | | |
|---|---|---|---|---|
| **Clusters** | **Number of cases** | **R** | **MB [km]** | **Y=aX+b** |
| **Coastal stations:** AKY, ATZ, SAL, LIM, CYC | 43 | 0.36 | -2.12 | 0.14x+0.31 |
| **Mountainous stations:** POT, GRA, EVO | 17 | 0.68 | -2.67 | 0.2x+0.11 |

A relevant sentence has been added to the revised manuscript regarding the effect of topography.

**14.** "In this study, the lidar data were analyzed using the KF method whenever the weather conditions were adequate, and the signal quality was sufficient for deriving high- quality backscatter vertical profiles". However, equation 1 is based on backscatter coefficient (?). Therefore, which is the reason why KF is used to compute LRs and extinction coefficients? Please provide more information and explain in the manuscript. If not needed, remove the KF/Raman sub- paragraph.

We thank the Reviewer for pointing this out. We do not use Raman retrievals of extinction and lidar ratio for the validation. We deleted the corresponding sentences to avoid confusion. The text was modified accordingly, and a clear explanation has been added.

**15.** Lines 152-160. Move to end of introduction, where the manuscript provides EARLINET implementation for passive sensors.

We accept the reviewer's suggestion. The sentences were adapted accordingly in the introduction section.

**16.** Lines 161-168. Non-relevant to the study.

We added these sentences here since even though the study mentioned validates a different satellite product, it does so using the EARLINET database and the methodology used in this work. We hence consider it an important asset for the overall understanding of the reader.

**17.** Section 2.1. Provide a table of the QA procedures applied to EARLINET.

The Single Calculus Chain (SCC) is the standard EARLINET tool to perform automatic and quality checked analysis of raw lidar data. All the operations implemented in SCC modules (HiRELPP, CloudScreen, ELPP, ELDA etc) are designed to preserve both the vertical and time resolution as high as possible. Some instrumental effects (like for example, dead-time correction, trigger-delay correction, overlap correction, atmospheric and electronic background subtraction, low- and high-range automatic signal glueing) are corrected following the recommendations provided by the EARLINET quality assurance program. The development of the SCC modules is continuing. (Additional info in detail: SCC products — Single Calculus Chain 5.2.0 documentation (cnr.it))

Following the Reviewer recommendation, a relevant table describing the QA procedures of EARLINET has been added in the manuscript. The text was modified accordingly.

**Table RC3-4.** QA procedures applied to EARLINET lidar measurements.

| SCC module | Procedure |
| --- | --- |
| HiRELPP | Corrections on the raw lidar signals. (dead-time correction, trigger-delay correction, overlap correction, atmospheric and electronic background subtraction, low & high-range automatic signal glueing) |
| CloudScreen | Clouds detection and screening on the pre-processed RCS timeseries. |
| ELPP | Corrections & transformations on the raw data before they can be used to derive the optical products at low temporal/spatial resolution. |
| ELDA | Retrieval of extinction & elastic/inelastic backscatter retrieval profiles |

**18.** Section 2.2. Provide a table of the QA procedures applied to S5P.

The TROPOMI/S5P pixel selection scheme and flags applied in the presented validation study, were made following the recommendations on the Product Readme File (PRF), Product User Manual (PUM) and Algorithm Theoretical Basis Document (ATBD) associated with the ALH product, all available on https://sentinels.copernicus.eu/web/sentinel/technical-guides/sentinel-5p/products-algorithms.

The product includes a quality assurance value (QA), which is a continuous quality descriptor ranging from 0 (no data) to 1 (full quality data) enabling end users to easily filter data for their own purpose. In order to avoid misinterpretation of the data quality, we exclude TROPOMI pixels associated with a

"qa_value" below 0.5. This removes very cloudy scenes, snow- or ice-covered scenes, and problematic retrievals. The ALH is very sensitive to cloud contamination and the height will be strongly biased towards the cloud height for partially clouded pixels. In addition, cloud flags are available from VIIRS and are strongly recommended to filter for residual clouds (See also our response to RC2). Pixels with an associated negative AI are excluded; hence only desert dust, biomass burning aerosol and volcanic ash aerosols – i.e. absorbing aerosols – remain in the dataset. No attempt is made to retrieve properties of non-absorbing aerosols. A sun-glint mask is also available to screen sun-glint regions, which are not filtered beforehand. Pixels covered by snow or ice are excluded. Oceanic pixels for which the viewing geometry is such that sunglint is to be expected are identified and excluded from analysis.

The QA procedures applied to TROPOMI retrievals are provided in detail within the text.

**19.** "The profiles from different types of lidar instruments have to be interpreted in terms of their ALH profile parameter (e.g. height of the assumed single aerosol layer) in a consistent way to reduce mismatch errors due to the significant different horizontal sensitivity between TROPOMI and lidar measurements" -> not clear, please re-write.

The sentence has been re-phrased accordingly.

**20.** A better justification on the selection of the collocation criteria should be used. For instance, in Pappalardo et al., 2014, the key connection link was based on meteorology. In Gkikas et al., 2022, more strict criteria are applied, while the air masses to be compared are related to the station measurements in-time through trajectories. The +-4 hours times window raises questions on the homogeneity of the air masses that are compared, for the atmospheric scene may have well changed in an eight-hours' time-window. How did the authors check/ensured that the cases were homogeneous enough to compare?

We agree with the Reviewer that the selection of collocation thresholds is critical point and raises questions on the homogeneity of the plumes under study. The temporal/spatial thresholds define the number of cases that are used in the analysis and the number of cases is critical to assess the performance of TROPOMI/S5P. In the revised manuscript we provide a better and more organized explanation regarding the selection of criteria.

The check of the homogeneity of the aerosol scenes considered in the comparisons we investigated for each scene the following:

1. Additional information about the temporal and spatial evolution of the detected aerosol layers is obtained from the forecast maps of dust load in the atmosphere, made by the regional BSC-DREAM8b dust forecast model.
2. Data from sun photometers belonging to the AERONET network were used complementary in order to assess the persistence and time evolution of the observed scenes.
3. Furthermore, to identify the dispersion of aerosol plume spatially and temporally for each case separately, True color images from different passive sensors such as MODIS/Aqua-Terra and VIIRS /Suomi-NPP were used extensively for each of the selected case.

For the ground-based lidar observations the aerosol backscatter profiles are derived considering a time window of ±2hour around the satellite overpass. Nevertheless, this temporal criterion has been relaxed or shifted in few cases to include selected cases related to an intense dust event persistent in the atmospheric scene, increasing the matched pairs with the TROPOMI ALH pixels. In most of cases the time difference between the mean averaged lidar profiles and S5P overpass vary from 1 to 3 hours (See the Figure RC3-1 below). Overall, 60% of the cases correspond to less than 1h, the 46.0% below 2 hours and only 2 cases are close to 4 hours. We should clarify that the long-time window was chosen to accept two cases of lidar measurements associated with intense dust transport episodes.

[Figure]

[Figure]

**Figure RC3-1**. Distribution of time difference between S5P overpass and lidar measurement average time for the selected collocated cases.

Our spatial collocation criterion is the common procedure applied in numerous studies related to the validation of satellite retrievals. The station must be the center of the circle area. Comparison of TROPOMI and EARLINET ALH values have been performed for a fixed maximum distance <150km Different radius distances thresholds in spatial scale have been investigated and the selected one is a compromise between the representativeness and the number of pixels that pass the QA thresholds within a search area. Finally the mean value of the satellite The mean of the ALH retrievals within the radius circle of 150km around the EARLINET stations is used for the comparisons and the corresponding standard deviation is a metric of the ALH variability within this search area typically within 1km (see figure RC3-2).

In the revised text, we are clarifying that the aforementioned data are used just as an indicator of the aerosol load in the surrounding area of the station synergistically with the Lidar observations.

**21.** Section 2.3 – number 4.: we use the lidar backscatter coefficient profiles mainly at 1064 nm (or 532 nm), analyzed by the SCC. Which is the reason that the authors did not use only 1064nm or 532nm and used in some cases the 1064nm and in other cases the 532nm. How did this selection affect the study, due to the different detection and scattering properties of aerosol in 532nm and 1064nm?

We used 1064nm and when this was not available only then 532nm. Applying our methodology (discussed in detail through the manuscript) to different wavelengths, we conclude that there is no significant dependence on the choice of wavelength ranging 20-80m. In the revised text, the authors make clear the reason for choosing the specific wavelength at 1064nm.

**22.** Does S5P provide pixels "aerosol-free"? If yes, are these cases included or removed from the analysis?

S5P provides only pixel retrievals containing information about the presence of aerosol and clouds. The findings of the present study refer to the presence of absorbing particles in the atmosphere. Pixels with an associated negative AI are excluded; hence only desert dust, biomass burning aerosol and volcanic ash aerosols – i.e. absorbing aerosols – remain in the dataset.

**23.** "… parameters such as the layer base (ZBASE), layer top (ZTOP), layer thickness (LTH) and center of mass (ZCOM), can be also calculated from the lidar signals…" Please provide the values, including the number of layers.

We removed this sentence from the manuscript, as it might be confusing for the reader. We do not use the geometrical features of the layers in the comparisons, since the retrieved ALH from TROPOMI corresponds to an effective aerosol weighted height and we compare it with a backscatter coefficient weighted height from the lidar (See Eq.1). We only use the layering tool described in order to

distinguish between single or multilayer structures in the comparisons. See our response to comment #26.

**24.** "The total available dataset is on the small side but suitable for the comparison study and general representativeness of the TROPOMI ALH product." Although the dataset may be sufficient to reach some conclusions, in my opinion, strong statements on the performance of S5P should be avoided or used with caution when the dataset is characterized by a low availability of intercomparison cases.

We accept the reviewer's opinion and modify the text accordingly where we deem it necessary.

**25.** Line 317: add relative bias.

We have computed the mean Relative Bias (RB) and included in the table of comparison statistics between TROPOMI and EARLINET.

**26. Major comments:**

A major question in algorithms is not merely "how good are the QA datasets", but also "how but are the non-QA datasets". In this case the question is which is the performance of "land only"? The authors should provide some metrics on this major category, as done in the "ocean only" and "ocean-land" categories and comment/discuss although the dataset is not extensive

Taking into account the reviewer's suggestions, we made an additional comparison analysis between S5P and EARLINET, assuming only land pixels. The results are also included in the revised version of the manuscript, replacing the previous data set that reported as combined land and sea. In addition, the number of coincidences with EARLINET stations has been extended from 34 to 63 cases, covering the time period from May 2018 till July 2022. The S5P data used overall in the study are produced under the ALH algorithm versions 01.03.00 to 02.03.01. In the study the authors use the maximum number of available EARLINET cases, to avoid any possible selection effect resulting from a poor sample of correlative cases, when strict collocation filters are applied.

The results of the comparison between the TROPOMI and EARLINET-derived aerosol heights for identified collocated cases, are shown in the **Figure RC3-2** (Top panel). The **Figure RC3-2** depicts the overall scatterplot between ground-based lidar and S5P ALH retrievals over (left) ocean and (right) land, as a function of the UV aerosol index (354/388nm), indicating the presence of absorbing aerosols in the atmospheric scene. What is immediately clear is that the lidar $ALH_{bsc}$ values are higher than the TROPOMI ALH over land. With an average difference of -2.28 km, median difference of -1.92 km and a standard deviation of 1.17 km, the retrieved ALH from TROPOMI over land is reported to be systematically lower than the Lidar $ALH_{bsc}$. The, as expected, good comparisons are acquired when only ocean pixels are included, which results in a mean difference of -0.51 km, a median difference of –0.53 km and a standard deviation of 0.77 km. There are several cases over the ocean where S5P ALH is significantly higher than the lidar retrievals. The results are very skewed over land, with very large negative values dominating the average. This can be seen also in **Figure RC3-2** (bottom panel) which presents histogram plots of absolute differences. The error bars shown in the scatterplots correspond to the standard deviation of the mean value including all the pixels in the selected radius around the station. The associated comparison statistics are summarized in **Table RC3-5**.

[Figure]

**Figure RC3-2**. (Top panel) Scatterplot of TROPOMI ALH retrievals against lidar calculated ALH for over ocean and land surface and (Bottom panel) Histogram of differences between lidar weighted backscatter height and TROPOMI ALH from collocated data between May 2018 and July 2022

**Table RC3-5.** Statistical metrics from the validation between S5P and EARLINET sites.

| | Number of cases | R | MB [km] | RB [%] | Y=aX+b |
|---|---|---|---|---|---|
| Over ocean | 63 | 0.82 | -0.51±0.77 | -14.73 | 0.69x+0.41 |
| Over land | 60 | 0.51 | -2.28±1.17 | -73.36 | 0.18x+0.21 |

The original **Figure 2** and **Tables 2, 3** in the original manuscript, have been modified accordingly to include additional statistics as mentioned already in the revised manuscript. The discussion is also modified accordingly following the results and discussion above.

Due to the aforementioned effects arising from the surface type (ocean/land), TROPOMI representativeness and performance is higher considering only ocean pixels, while TROPOMI performance over the land is characterized by extreme lower value of mean bias and at the same time by moderate correlation coefficient than in the case of ocean cases.

Moreover, the authors should provide a separate analysis for the cases that only one layer was detected by EARLINET, and for the cases where two-or-more layers were detected by EARLINET (two clusters), for the structure of atmospheric aerosol layers is a challenging task for a passive sensor affecting ALH, as also discussed in the section of S5P.

We agree with the Reviewer that the structure of atmospheric aerosol layers is a challenging task for a passive sensor affecting ALH. Following the recommendation, we provide a separate analysis for the cases that only one layer was detected and for the cases where two-or-more layers were detected by EARLINET. The lidar dataset has been processed by an automatic geometrical features detection algorithm using the Wavelet covariance transform technique (**See in Appendix A2**). Several cases where distinctly separated elevated and boundary layer aerosols are present in the same scene - satellite

product derives one layer. Tables **RC3-6 & RC3-7,** summarize the comparison statistics between EARLINET & S5P under the different number of detected layers in the atmospheric scene for ocean and land pixels respectively.

Overall, lidar data reveal the presence of a single aerosol layer in 57.1% (N=36) of sample cases and a multilayer structure (two-or-more layers) the 42.9% (N=27) of the total sample cases. Regarding the results, the mean bias of the difference [TROPOMI-Lidar] seems to become larger when multilayers exist in the atmospheric scene, but we cannot say with certainty that it is the general rule due to the use of a limited validation dataset. Overall, among all the cases the best performance of TROPOMI is recorded in case of single well-developed dust layers.

However, more research is needed to substantiate the observations and make conclusive quantitative statements. Particularly, more cases should be included in future retrieval experiments to improve the statistics and see whether our findings are robust.

(The Tables are not included in the revised manuscript, however a relevant sentence added in the text-Section 3.1)

**Table RC3-6.** Statistical metrics for the comparison S5P vs EARLINET over ocean for different number of layers.

| TROPOMI pixels over ocean | | | | |
|---|---|---|---|---|
| **Number of layers** | **Number of cases** | **R** | **MB [km]** | **Y=aX+b** |
| 1 | 36 | 0.8 | -0.46 | 0.71x+0.38 |
| > 2 | 27 | 0.85 | -0.55 | 0.67x+0.47 |

**Table RC3-7.** Statistical metrics for the comparison S5P vs EARLINET over land for different number of layers.

| TROPOMI pixels over land | | | | |
|---|---|---|---|---|
| **Number of layers** | **Number of cases** | **R** | **MB [km]** | **Y=aX+b** |
| 1 | 35 | 0.68 | -2.2 | 0.24x+0.047 |
| > 2 | 25 | 0.25 | -2.38 | 0.09x+0.51 |

Finally, frequently, it is mentioned that it is a challenge of S5P to detect ALH over land due to surface albedo. As an evaluation study, I would expect to provide through a surface- type database an assessment on the effect of different surface types. For instance, it is mentioned: "Many factors can play a role in this apparent disagreement between TROPOMI retrievals over land and sea including that high surface albedos negatively influence the ALH, biasing the ALH towards the surface." Though this is known, and frequently mentioned in the manuscript, as a validation study this is something that should be more extensively addressed.

We thank the reviewer for pointing out these aspects as they play an important role in our paper's analysis and results. The influence of the surface albedo is discussed, and the aspects of the TROPOMI ALH algorithm that will require future development efforts are highlighted.

It is true that a number of known data Quality Issues exist for S5P ALH product. Based on the TROPOMI ALH ATBD, it is known that high surface albedos negatively influence the ALH, biasing the ALH towards the surface. In general, the ALH over (dark) oceans is considered reliable to within the requirement of 1000 m or 100 hPa. Over land, especially bright surfaces, the accuracy may be lower, and the use of the ALH product over bright surfaces like deserts is not advisable. A surface albedo climatology is used to provide a priori values for retrieval. The corresponding versions of the ALH algorithm for the data used in the study, implemented in the Level-2 processor does not fit the surface albedo but keeps it fixed in the retrieval at climatological values. Sensitivity studies showed favorable results when fitting of the surface albedo was included in the retrieval procedure. A recent test with synthetic test cases covering a large number of surface albedo values showed that including the surface albedo in the optimal estimation fit considerably improves the ALH in most cases [see Section 4.6 in ATBD]. Based on these results, it is foreseen that the surface albedo will be included in the optimal estimation fit at least over land or for a bright (enough) surface. However, to include this, the NN needs to be retrained.

The effect of the surface albedo has been investigated through the sensitivity tests presented in detail in previous studies showing that an accurate ALH retrieval critically depends on the appropriate assumption of surface reflectance (Sanders and de Haan, 2016; Ding et al., 2016; Dubuisson et al., 2009).

We confirm through our results (see Figure RC3-2 and Table RC3-5) that there is a clear dependence of the satellite product on height due to the effect of ground reflectivity. The figure below shows the effect of the surface type on the TROPOMI ALH retrievals for a smoke scene over the Iberian Peninsula on 24 October 2020. We provide this case, to emphasize the issue related to the limitation of satellite measurements over land areas where the effect of the surface reflectance is dominant. The contrast observed between land and sea regarding the retrieval of the ALH product and the surface albedo values is obvious as can be seen from the color scale. The ALH retrievals are very clearly biased over land. It is evident that high surface albedo biases ALH low. The differences between lidar and TROPOMI increase with increasing surface albedo, consistent with the idea that the TROPOMI retrieval algorithm is more sensitive over dark surfaces and the error seems to increase with increasing surface albedo. This issue has also been studied and confirmed by previous studies (Griffin et al., 2020; Nanda et al., 2020).

[Figure]

**Figure RC3-3**. (Left) VIIRS/Suomi-NPP true color image, (middle) S5P ALH and (right) S5P surface albedo retrievals over Iberian on October 24, 2020, where an elevated smoke plume is detected.

A relevant discussion has been added in the revised manuscript (Section 4) presenting the effect of the surface albedo in the validation analysis.

The following references have been added in the revised manuscript:

Ding, S., Wang, J., and Xu, X.: Polarimetric remote sensing in oxygen A and B bands: sensitivity study and information content analysis for vertical profile of aerosols, Atmos. Meas. Tech., 9, 2077–2092, https://doi.org/10.5194/amt-9-2077-2016, 2016.

Dubuisson, P., Frouin, R., Dessailly, D., Duforêt, L., Léon, J.-F., Voss, K., and Antoine, D.: Estimating the altitude of aerosol plumes over the ocean from reflectance ratio measurements in the O2 A-band, Remote Sens. Environ., 113, 1899–1911, https://doi.org/10.1016/j.rse.2009.04.018, 2009

Nanda, S., de Graaf, M., Sneep, M., de Haan, J. F., Stammes, P., Sanders, A. F. J., Tuinder, O., Veefkind, J. P., and Levelt, P. F.: Error sources in the retrieval of aerosol information over bright surfaces from satellite measurements in the oxygen A band, Atmos. Meas. Tech., 11, 161–175,https://doi.org/10.5194/amt-11-161-2018, 2018.

**27.** Figure 2a: The datasets are high correlated, according to the results. Could the authors provide to the figure or in the manuscript, the correlation coefficients of the "dust"and "smoke" cases also separately?

Following the reviewer's suggestion, we performed an analysis separately for defined cluster (dust/smoke). A summary of the statistic metrics is shown in tables RC3-8, RC3-9 per defined cluster. Relevant phrases have been added in section 3 accordingly within the text, reflecting the findings.

We separated the desert dust and the biomass burning cases including in the validation dataset in order to investigate any possible impact of the different type in the results. A summary of the statistic

metrics is shown in tables below per defined cluster (dust/smoke). The identification of the type of aerosol plume for each case is done by using synergistic tools such as the HYSPLIT back-trajectories (Stein et al., 2015), the regional BSC-DREAM8b dust forecast model (Pérez et al., 2006; Basart et al., 2012) and columnar aerosol optical properties obtained from collocated AERONET (Holben et al., 1998) stations. In addition, when is available the information for depolarization ratio from the lidar measurements is also considered. A more detailed work related to aerosol typing is beyond of the scope of this study.

Most of the collocated cases included in the analysis are referred to homogeneous well-developed dust (N=57) and only eight (N=6) referring to smoke events, which may be more or less homogeneously distributed. However, it has to be taken into consideration the important factor related to the presented results that is the number of TROPOMI-EARLINET correlative cases used in the analysis. When only retrievals above land are considered, the total number of cases is reduced to 60. We see that the MB is increasing when elevated smoke layers are present in the atmospheric scene. S5P ALH seems to be more accurate for low altitude dust layers, very close to product requirements. Thin plumes and high-altitude plumes may be missed.

**Table RC3-8**. Collocated cluster cases over ocean with respect to the source origin (smoke & dust).

| TROPOMI pixels over ocean | | | | |
|---|---|---|---|---|
| **Source** | **N** | **R** | **MB [km]** | **Y=Ax+b** |
| Dust | 57 | 0.57 | -0.42 | 0.61·x+0.64 |
| Smoke | 6 | 0.98 | -1.28 | 0.77·x+0.20 |

**Table RC3-9.** Collocated cluster cases over land with respect to the source origin (smoke & dust).

| TROPOMI pixels over land | | | | |
|---|---|---|---|---|
| **Source** | **N** | **R** | **MB [km]** | **Y=Ax+b** |
| Dust | 55 | 0.31 | -2.04 | 0.18·x+0.22 |
| Smoke | 5 | 0.75 | -4.9 | 0.21·x+0.01 |

The Tables are not included in the revised manuscript, however a relevant sentence added in the text-Section 3.1.

**28.** Figure 5: "TROPOMI is in excellent agreement with the calculated $ALH_{bsc}$ from the lidar profile". However, the two layers are ~ 0.5 km retrieved differently. Please avoid so strong statement, not supported.

Accepted.

**29.** Line 514: In this part should also be discussed the effect of the 760m elevation of the EARLINET station in the intercomparison.

We thank the reviewer for pointing out this aspect as it plays an important role in our paper's analysis and results. It must be clearly understood by the reader and well-marked within the text. In the revised version, where the overlap region was also considered in our calculations for the lidar estimated ALH we also considered as the lowest height, the altitude above sea level of each station.

The text has been modified to include additional information related to the overlap height effect also considering the station altitude, as mentioned already in our response to the previous comment of the reviewer RC1.

**30.** "The statistical results show the ability of the TROPOMI instrument to detect aerosol layers under cloud-free atmospheric conditions with significant aerosol load, such as dust and smoke plumes". Following the cases presented, and discussed, avoid a so define statement. Moreover, the number of cases prohibits statements such as "Overall, our results testify that the TROPOMI product complies with the S5P mission requirements" and "This work confirms that the TROPOMI ALH product is within the required threshold accuracy and precision requirements of 1 km.".

Considering the reviewer's recommendation, the text was modified accordingly.